



# 1 Global spatially-distributed sectoral GDP map for disaster risk

# 2 analysis

Takeshi Shoji[1,2], Dai Yamazaki[1,2], Yuki Kita[2,3], Megumi Watanabe[2,4]
[1]Graduate School of Engineering, The University of Tokyo, Tokyo, 113-8656, Japan
[2]Institute of Industrial Science, The University of Tokyo, Tokyo, 153-8505, Japan
[3]Gaia Vision Inc., Tokyo, Japan
[4]LERMA, Observatoire de Paris, Paris, 75014, France
*Correspondence to*: Takeshi Shoji (tshoji@rainbow.iis.u-tokyo.ac.jp)
**Abstract.** Global risk assessments of economic losses by natural disasters while considering various land uses is essential.
However, sector-specific, high-resolution pixel-level economic data are not yet available globally to assess exposure to local
disasters such as floods. In this study, we employed new land-use data to construct global, spatially distributed map of
sector-specific gross domestic product (GDP). We developed three global GDP maps in 2010, 2015, and 2020 for service,
industry, and agriculture sector, with 30 arcsec resolution. Firstly, we found that the spatial relationship between the
distribution of industrial GDP and urban areas, where the service GDP is highly concentrated, varies across countries. For
example, in the United States, industrial GDP is widely dispersed regardless of urban areas, whereas in India, industrial GDP
is concentrated in proximity to urban areas. Secondly, we evaluated the GDP map by subnational regional statistics of
Thailand, where validation data are accessible. Traditional GDP maps relying solely on population distribution exhibited
63.0% relative error of the sectoral GDP in each subnational region to regional statistical data, which the new sector-specific
GDP map reduced to 26.2%. Subsequently, we assessed the map in conjunction with sector-level business interruption (BI)
losses resulting from river flooding. Our estimation of sector-level losses revealed that the sectoral ratio to the total loss
varied significantly depending on the spatial distribution of flood hazards. The estimated total loss became closer to the
reported value when the new GDP map was used, while sectoral ratios of losses still had some differences from the reported
ratios suggesting the need for further improving the procedures of loss-estimation models. These global sectoral GDP maps
(SectGDP30) are available at https://doi.org/10.5281/zenodo.13991673 (Shoji et al., 2024).

## 25 1 Introduction

In recent years, as natural disasters have become more frequent and found throughout the world (IPCC, 2012), global spatial
data including land use and socioeconomic information have become essential for estimating the extent of disaster damage
and losses. With the increasing frequency and impact of localized natural disasters such as floods, high-resolution data

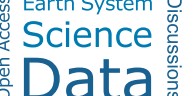

capturing the spatial distribution of socioeconomic factors are essential. However, socioeconomic data published by
international organizations such as the World Bank are often available only at the national or large municipal level. At the
research level, economic data at the municipal level have been studied (Wenz et al., 2023); however, obtaining grid-level
data at a resolution of several kilometers has been still challenging.

For example, as for the impact-assessment of flood disasters, researchers have undertaken a series of studies by spatially
calculating the amount of asset quantity and production activity overlapped with inundated areas, leveraging global maps.
Achieving this necessitates the downscaling of national-level data of economic activity, mainly gross domestic product
(GDP), to finer subnational or grid-based levels. This type of product by downscaling GDP is called a "spatially distributed
GDP map". This downscaling practice typically relies on gridded population data (Tanoue et al., 2021; Willner et al., 2018).
Alternatively, it has involved the assembly and interpolation of available subnational statistics (Duan et al., 2022; Kummu et
al., 2018) or the assumption that average building heights correlate with economic activity intensity (Taguchi et al., 2022).

While these studies estimated the total amount of economic losses without considering the difference between sectors, the
sector-classified economic losses also need to be estimated because indirect economic losses, such as global supply chain
impact caused by the stoppage of production activity (Willner et al., 2018), can vary significantly depending upon the sector
directly affected by the flood (Sieg et al., 2019). However, spatial data of sectors by downscaling national-level data have
been lacking. Consequently, in the context of global studies, the estimation of sector-specific losses was achieved by
extrapolating the values of sectoral occupation fractions within urban area grids, as reported in the European Union, to other
regions (Alfieri et al., 2016; Dottori et al., 2018). Alternatively, it is assumed that specific groups of sectors experience
uniform damage ratios (Willner et al., 2018; Tanoue et al., 2020). These methods did not consider the different spatial
accumulation between each sector and each region, which could lead to the misestimation of sector-classified losses
(Jongman et al., 2012; Willner et al., 2018).

The dearth of global spatial data of the economic sector arises from the absence of worldwide maps with comprehensive land
use categorizations (Wenz and Willner, 2022). While regional maps provide sectoral land use classifications, including
commercial and industrial areas within urban regions (e.g., European Environmental Agency, 2017; Theobald, 2014; De
Moel H et al., 2014; MLIT 2021), these classifications are conspicuously absent from global maps (e.g., Bontemps et al.,
2011; Esch et al., 2017). Here we focused on the recent emergence of a global land use map featuring detailed urban area
classifications (Pesaresi and Politis, 2022). This development is made possible by the application of machine learning
techniques that extrapolate relationships between satellite observations and actual land uses, a methodology initially
established by the data in the European Union and the United States (European Environmental Agency, 2017; Theobald,
2014) and subsequently extended to a global scale. Although this dataset facilitates a comprehensive consideration of



detailed land-use patterns within urban areas worldwide, no study has yet integrated this dataset with socioeconomic data.
Such integration holds the potential to pioneer a novel approach to estimating natural disaster damage accurately with
sectoral classifications.

The objective of this study is to leverage a recently available global detailed land use map dataset to construct a spatially
distributed sectoral GDP map. The accuracy of the distribution of economic sectors within this newly developed spatially
distributed GDP map is evaluated using data from Thailand. Validation is achieved by scrutinizing the consistency of
subnational statistics within Thailand. Furthermore, to discuss the applicability of the new GDP map for practical economic
loss estimation, this study examines the estimation of business interruption losses incurred due to a flood event in Thailand
and compares these estimations with reported values. The reason for choosing Thailand as a target of validation was that this
country has both sectoral subnational GDP statistics and the reported values of sectoral economic losses caused by the
historical event while most countries do not have nor publish those types of data.

## 74 2 Methods

### 75 2.1 Spatially distributed sectoral GDP map

The spatially distributed sectoral GDP map was created in two steps (Figure 1). First, we created a global sectoral land use
fraction map at a spatial resolution of 30 arcsec, and combined satellite products to classify three sectors: the service,
industry, and agricultural sectors. Then, country GDP data classified according to these sectors were distributed spatially on
the corresponding sectoral area fractions in the global sectoral land use fraction map. The List of the datasets used in this
method is shown in Table 1.




**Figure 1: Flowchart of (top) data processing and (bottom) creation of spatial distributed gross domestic product (GDP) maps of Thailand for the (a) service, (b) industrial, and (c) agricultural sectors.**




| Data | Format | Datatype | Values range | Spatial resolution | Temporal resolution | Data source, Reference |
|---|---|---|---|---|---|---|
| Built up surface area<br>Non-residential surface area | Raster | UInt16 | 0-10000 | 100m | five years interval (1975-2020) | Global Human Settlement Layer (Pesaresi and Politis, 2022) |
| Crop land area | Raster | Boolean | 0,1 (0 - no croplands, 1 - croplands) | 0.9 arcsec | five years interval (2003-2019) | Potapov et al., 2022 |
| Population count | Raster | Float64 | 0-Inf | 30arcsec | five years interval (1975-2020) | Global Human Settlement Layer (Pesaresi and Politis, 2022) |
| City area polygons | Vector (Polygon) | - | - | - | - | Global Rural-Urban Mapping Project v1 (CIESIN, 2011) |
| Administrative units | Vector (Polygon) | - | - | - | - | GADM 4.1 (2023) Level 1 Layer |

**Table 1:List of the datasets used in this study.**

In the first step, we used land use classification maps from satellite products to produce a global sectoral land use fraction map. We generated a sectoral land use fraction map classified into three sectors (service, industry, and agriculture) and three land use type maps with different spatial resolutions: residential (RES), non-residential (NRES), and cropland (CROP). To distinguish RES and NRES areas, we used Global Human Settlement Layer (GHSL) (Pesaresi and Politis, 2022) built-up surface (R2022) data. This layer has $100 \times 100$ m resolution; each pixel has a value of 0-10,000 m2 and residential or non-residential areas may be present within one pixel. For CROP area, we used the global map of cropland extent (Potapov et al., 2022), provided by Global Land Analysis & Discovery, which has a global spatial resolution of 0.9 arcsec. Maps with the three classes were resampled and combined into a single global sectoral land use (residential, non-residential, and cropland) fraction map at 30-arcsec resolution.

First, we upscaled the land use maps and simultaneously converted the value of each pixel in both maps into the sectoral fraction within one pixel. In each pixel, RES and NRES had values of 0–10000 m2 and CROP had a value of 0 or 1 (not cropland or cropland). We upscaled the land use maps to 30-arcsec resolution from RES and NRES at a resolution of $100 \times 100$ m and CROP at a resolution of 0.9 arcsec using the GDAL averaging method (GDAL/OGR contributors. 2024). Using the 30-arcsec maps, we calculated the area attributed to each land use type in one pixel with a size of $1 \times 1$ arcsec and obtained land use fractions for each pixel. Because RES/NRES and CROP had different data sources, the total of the three land use type fractions was greater than one in some pixels. Therefore, we assumed that the CROP fraction could fill only areas that were not designated as RES or NRES. Under this assumption, we modified the CROP fraction in each pixel as follows:

$$MCROP_i = min\left(CROP_i, \left(1 - RES_i - NRES_i\right)\right) \tag{1}$$

where $MCROP_i$ is the modified CROP fraction in pixel i, $CROP_i$ is the original CROP fraction, $RES_i$ is the RES fraction, and $NRES_i$ is the NRES fraction.



After this modification, RES, NRES, and MCROP were considered to represent the service, industrial, and agricultural land
use sectors, respectively.

In the second step, we spatially distributed the country-level GDP onto the global sectoral land use fraction map generated in
the first step. We used GDP data published by the World Bank (2023), which includes both yearly GDP values and their
sectoral ratios for the service, industrial, and agricultural sectors. For industrial and agricultural GDP, we assumed that the
sectoral GDP per area was the same in all the areas of that sector within each country; thus, the industrial and agricultural
GDP were distributed only in proportion to the sectoral area fractions of each pixel, with a size of 30 × 30 arcsec.

To create a spatially distributed sectoral GDP map, we distributed the sectoral GDP into each sectoral land use area, in each
country by multiplying the distributed sectoral GDP per pixel by the sectoral area fraction in each pixel. At this step, we
assumed that the distributed sectoral GDP per pixel was the same only within the same country and the same sector. Thus,
the distribution was performed for each country and each sector, as follows:
$$SGDP\ per\ pixel_{country,s} = \frac{TtlSGDP_{country,s}}{\sum_{i=1}^{n} SA\ fraction_{i,s}}$$ (2)
$$SGDP_{country,i,s} = SGDP\ per\ pixel_{country,s} \times SA\ fraction_{i,s}$$ (3)
where $SGDP\ per\ pixel_{country,s}$ is the sectoral GDP per pixel of sector s in the country, $TtlSGDP_{country,s}$ is the total sectoral
GDP of sector s in the country, $SA\ fraction_{i,s}$ is the sectoral area fraction of sector s in pixel i, n is the total number of pixels
in the country, and $SGDP_{country,i,s}$ is the distributed sectoral GDP of sector s in pixel i in the country.

For the service GDP distribution, the activity level in each service sector area depends strongly on the number of people
living near that area and using services (Morikawa, 2011). Therefore, we considered the city effect only for the service
sector. As an appropriate scale for counting the number of neighbors using the services of a specific area, the grid-scale
population (e.g., 30-arcsec resolution, approximately 1 × 1 km per pixel) is too fine to describe a realistic number of users
because many people often travel further than 1 km by car or public transportation. Country and district scales are too broad
to reflect the intensity of demand of each area accurately (Ciccone and Hall, 1996). Additionally, population density
corresponds more strongly to economic activity than to population counts (Ciccone and Hall, 1996; IMF, 2019). Therefore,
we used city-scale population density information for the service sector GDP distribution (City Effect, Fig. 1).

The service GDP was distributed only in pixels within cities and the amount of distributed GDP was proportional to the
population density of the city where the pixel is located. To detect pixels included in cities, we used the global city polygon
dataset provided by Global Rural-Urban Mapping Project (GRUMP) v1 (CIESIN, 2011). To calculate the population density





of each city, we used the global gridded population map provided by GHSL population grid (R2023; Pesaresi and Politis,
2022). For the distribution of service sector GDP, we first masked out the fractions of the service sector in pixels that did not
belong to any city detected using the global city polygon dataset. We distributed GDP into only pixels that belonged to cities,
and we assumed that the GDP per area was the same in one city and that the amount of gridded GDP was in proportion to the
service sector fraction of each pixel. This calculation was performed as follows:
$$ServGDP\ per\ city_{country,city} = \frac{PD_{city}}{\sum PD_{city}} \times TtlServGDP_{country} \tag{4}$$
$$ServGDP\ per\ pixel_{country,city} = \frac{SectGDP\ per\ city_{country,city}}{\sum_{i=1}^{k} ServArea\ fraction_i} \tag{5}$$
$$ServGDP_{country,city,i} = ServGDP\ per\ pixel_{country,city} \times ServArea\ fraction_i \tag{6}$$

where $ServGDP$ is the service sector GDP, $PD_{city}$ is the population density of the city, $TtlServGDP_{country}$ is the total
amount of service sector GDP of the country, and $ServArea\ fraction_i$ is the fraction of service sector area in pixel i.

## 2.2 Comparison of GDP distribution methods

We created three types of spatial distributed GDP map: population-based (PB), sector-based (SB), and sector-based with City
Effect (SBCE). The PB map was generated by downscaling the country GDP only in proportion to the gridded population
count into a 30-arcsec map. The SB map was generated for each sectoral area and sectoral GDP per area, assuming that the
sectoral GDP per area is the same within each country. The SBCE map was generated by considering the city-scale
population density effect (City Effect) mentioned above, only for the service GDP distribution. The GDP of the industrial
and agricultural sector in the SB map and the SBCE map were distributed using the same method.

## 3 Results

We developed three GDP maps for service, industry, and agriculture sectors in 2010, 2015, and 2020. We excluded other
years because of the low coverage of national GDP statistics in the World Bank data. The developed sectoral GDP maps are
shown in Fig. 2 (a), (b), and (c). Additionally, to clarify the difference of spatial distribution among sectors, we showed (d)
the map of the largest GDP sector in each grid in the world and (e) around Thailand as an example. Although the GDP maps
were produced with a spatial resolution of 30 arcsec, these maps in Fig. 2 show the aggregated maps into 0.5 degree. These
GDP maps are those called SBCE in the Methods. Both maps of the service and industry sector showed the same shape of
extents which have each sector GDP. Meanwhile, the GDP accumulation into the center of the economic activity was
different between them. Looking at the east part of the United States, while industry GDP was scattered evenly in a wide
area, service GDP accumulated intensely in some centers of cities and other areas have much smaller GDP in those places.
This tendency was not the case with other countries. In countries such as India and Iran, the industry GDP was more
concentrated in some specific areas than the service GDP. As for the agriculture GDP, compared to maps of those two
sectors, the GDP was spread to a much wider area with less concentration in specific areas. Even with this different
characteristic, the agriculture GDP was basically distributed aligning with the other two sectors' GDP. When we look at the
map around Thailand (Fig. 2 (e)), we can see the different distribution between each sector. While the service GDP (blue)
dominated in the Bangkok area, the industry GDP mainly dominated in the eastern area, next to the Bangkok area. The
sectoral GDP map of this study showed such heterogeneity of each sector on a local scale within one country.

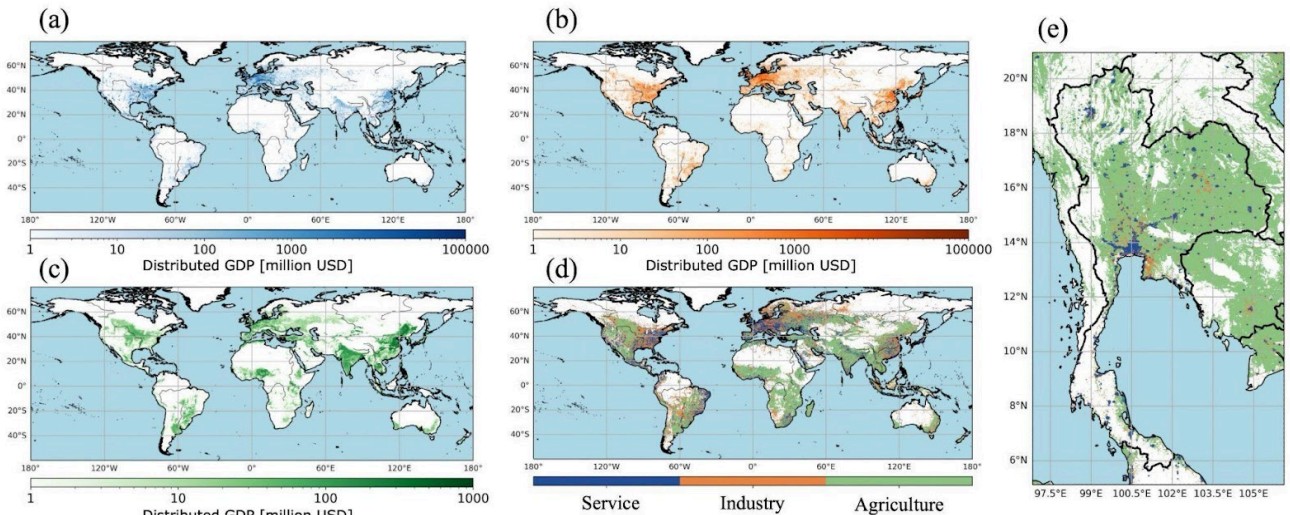

**Figure 2: The sectoral GDP maps of (a) service sector, (b) industry sector, (c) agricultural sector, (d) the map of the largest GDP**
**sector in each grid of 0.5 x 0.5 degree, and (e) the same map around Thailand.**

We validated this different distribution of each sector's GDP using subnational sectoral GDP statistics of Thailand in 2009
provided by the Thailand government (NESDC, 2016) as reference data. We spatially aggregated the GDP map into seven
districts corresponding to the statistics classification: Northeastern, Northern, Southern, Eastern, Western, Central, and
Bangkok & Vicinity (Fig. 3). This aggregation was performed using the administrative area polygon dataset obtained by
GADM 4.1 (2023) and its correspondence with the district definition in the statistics. The spatially aggregated GDP of each
sector in each district of the three maps (PB, SB, SBCE) and the Thailand government statistical values (Reference) are
shown in Fig. 3. The population-based map had no information on sectoral differences among districts; therefore, the
sectoral ratio of the gridded GDP value was assumed to match that of the entire country in all pixels and districts, following
the practice of previous studies (Willner et al., 2018; Tanoue et al., 2020). As an index of consistency of the three maps with



Reference, we calculated average relative errors (ARE) of the aggregated district GDP in each map to Reference, on an
average of all seven districts, as follows:

$$ARE[\%] \;=\; \frac{1}{7}\sum_{k=1}^{7}\left|\frac{Regional\,GDP_k - Ref_k}{Ref_k}\right| \times 100 \qquad (7)$$

where k is the number of each district shown in Fig. 3.
The AREs for the total GDP values of the PB, SB, and SBCE maps were 63.0%, 50.0%, and 26.2%, respectively. These
AREs consisted of errors of each sector in each district. For the service sector, the AREs were 50.3%, 69.2%, and 38.6%,
respectively, in the PB, SB, and SBCE map. The largest service GDP was seen in Bangkok & Vicinity in Reference. While it
was seen in the same district in the SBCE map, the different district (Southeastern) had the largest in the other two maps (PB
and SB). This result meant the SBCE showed better consistency with Reference than PB and even SB. This indicated that
solely using the residential fraction map was not enough to express the spatial distribution of service GDP and the city-scale
population density could help to reproduce the actual GDP distribution.

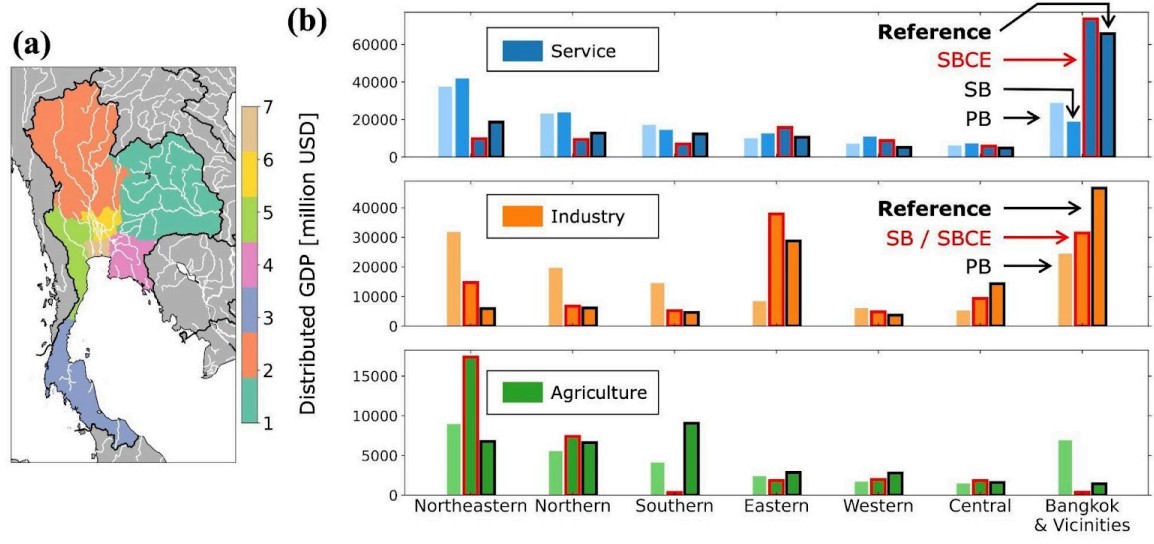


**Figure 3: (a) The seven districts of Thailand (1, Northeastern; 2, Northern; 3, Southern; 4, Eastern; 5, Western; 6, Central; 7, Bangkok and Vicinities). (b) Distributed sectoral GDP of subnational even districts in Thailand in 2009, obtained from the population-based (PB), sectoral-based with city effect (SBCE) maps and statistical values from the government of Thailand (Reference).**


For the industrial sector, the AREs were 159% and 42.7% in the PB and SB/SBCE map, showing the PB map had a marked
inconsistency to Reference. On the other hand, SB/SBCE maps could express the large industry GDP in districts such as





Eastern and Central. This indicated that the accumulation of non-residential fraction, which was hypothetically assumed to
correspond to industry GDP in this study, corresponded well with the distribution of industry sector activities.

Conversely, for the agriculture sector, which was spatially distributed using the same method as for the industrial sector,
none of the three maps could show the largest agriculture GDP in the Southern district. The SB/SBCE map showed an
overestimation in Northeastern and underestimation in Southern and Bangkok & Vicinity. This indicated that the cropland
fraction map used in the Method could not express the intense accumulation of agriculture GDP. The cropland map used in
this study has no information on crop types; thus, the productivity of individual crop types was ignored for each district in
Thailand. For example, the Northeastern district produces mainly rice with low land productivity, whereas the Southern
district produces natural rubber and palm oil (Inoue, 2010). This heterogeneity of "production in monetary unit per area" was
not considered in this study, which probably led to the low improvement of GDP distribution accuracy in the SB/SBCE map.
**4 Discussion**
**4.1 Business interruption loss estimation for the 2011 Thailand flood**
To assess how the improvement of the GDP map affects the result of flood loss estimation, an additional analysis of
estimating business interruption losses resulting from the actual flood event in Thailand in 2011 by the new sectoral GDP
map was conducted to assess how the improvement of the GDP map affects the result of flood loss estimation. Following
established definitions of economic losses from prior studies (Tanoue et al., 2020; Rose, 2004), economic impacts can be
categorized into three main types: damage, direct economic loss, and indirect economic loss. This additional analysis focused
exclusively on estimating BI loss among these three economic impacts due to the lack of information necessary for the
estimation of the other components.

To calculate BI loss, we prepared hazard, exposure, and vulnerability data. As the hazard, we used two inundation period
maps of the target event in Thailand, based on simulation and satellite observations. The simulation-based inundation period
map was generated using the Catchment-based Macro-scale Floodplain (CaMa-Flood) global riverine inundation model
(Yamazaki et al., 2011). To obtain an inundation map based on the simulation by CaMa-Flood, CaMa-Flood used daily
runoff data generated by a reduced-bias meteorological forcing dataset at 15-arcmin resolution, and S14FD-Reanalysis data
(Iizumi et al., 2017) to simulate the daily inundation depth at 15-min resolution. Because S14FD is a bias-corrected dataset,
we used daily inundation depth values without bias correction, such that the inundation period may be calculated directly
from the daily inundation depth (Taguchi et al., 2022). Then, we downscaled the 15-arcmin daily inundation depth to
30-arcsec resolution and calculated the inundation period as the number of days in which the inundation depth exceeded 0.5
m in each pixel. We also used an inundation period map based on Terra/Moderate Resolution Imaging Spectroradiometer
(MODIS) images, which is publicly available on the Global Flood Database (Tellman et al., 2021). We referred to the former
hazard map as "CaMa-Flood" and the latter map as "MODIS" in this study. The days between August and December in 2011
were only counted as inundation days for matching the inundation period by CaMa-Flood simulation and that by MODIS
observation, which started from August and ended around the end of December. The inundation period maps of CaMa-Flood
and MODIS are shown in Fig. 4.

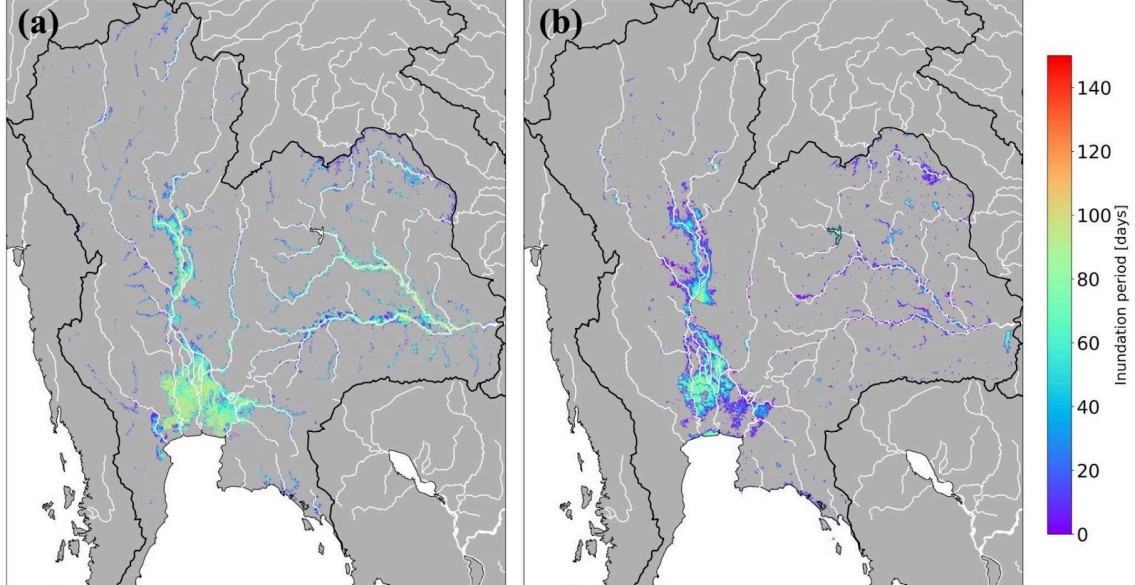

**Figure 4: Spatial distribution of the inundation period of the 2011 Thailand flood, obtained from (a) Catchment-based Macro-scale**
**Floodplain (CaMa-Flood) simulation and (b) Moderate Resolution Imaging Spectroradiometer (MODIS) observation data.**

As exposure, we used two spatial distributed GDP maps at 30-arcsec resolution for comparison, the population-based map
(PB) and the sector-based map with CE (SBCE). As a vulnerability, we considered a recovery coefficient, which decided the
ratio of the length of recovery period which is required until business restart to the inundation period. This value reflects the
system vulnerability of the city. We used 2 as a recovery coefficient, which was used in previous study on a global scale
(Taguchi et al., 2022). As for the recovery period as vulnerability, we used the method of Tanoue et al. (2020). The recovery
period $RP_i$, when the production in a pixel is assumed to have recovered linearly from zero at the end of the flood period to
the same level of production before the flood, was obtained by multiplying the inundation period by a coefficient (= 2 in this
study). Thus, the recovery period was assumed to take twice as long as the inundation period. Finally, BI loss was estimated
by the method described by Tanoue et al. (2020), as follows:
$$BI\ loss = \sum_{i=1}^{N}\sum_{s}^{3}\left\{(IP_i + \frac{RP_i}{2})\times\frac{AGDP_{i,s}}{Nd}\right\} \tag{8}$$





where *i*, *N*, and *s* are the pixel number, total number of pixels in the inundated area, and sector number (1 = service, 2 =
industry, and 3 = agriculture), respectively; $IP_i$, $RP_i$, $AGDP_{i, s}$, and $Nd$ are the inundation period, recovery period at pixel *i*,
annual GDP of pixel *i* and sector *s*, and the number of days in a year.
And we obtained the total BI losses by summing BI losses of all the grids in the target area.

The results of the BI loss estimation were shown in Fig. 5. We compared the calculated BI losses with the actual economic
loss reported in the PDNA (The World Bank, 2011). In this report, both damage and loss were estimated. Damage is due to
the destruction of physical assets and loss is caused by foregone production and income and higher expenditures in the
definition in the report. This means that the loss in the report included both business interruption loss and other additional
expenditures and costs. Because there was not any other reported loss which only focused on BI loss, we compared with the
loss, including other components, in this report.

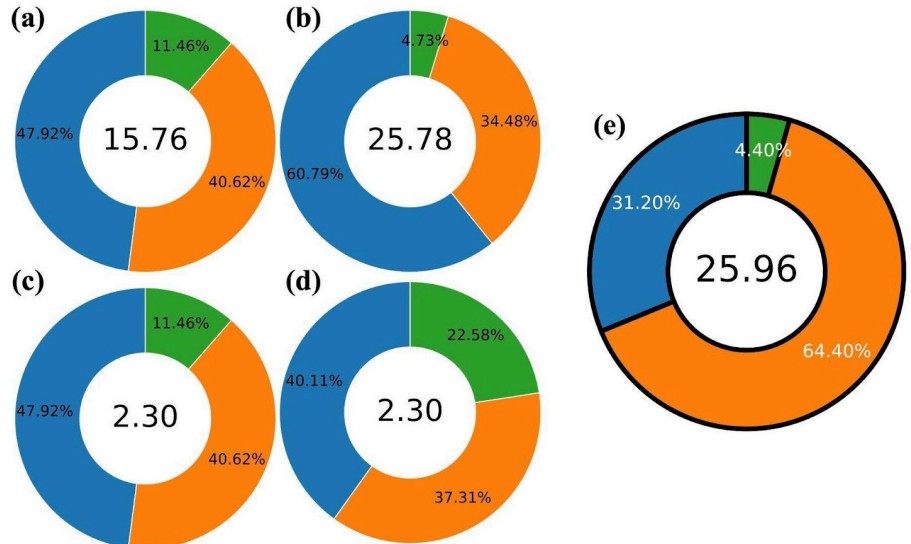

BI loss [billion USD, current value in 2011]


**Figure 5: Business interruption losses (USD billion, current value in 2011) due to the 2011 Thailand flood, estimated by combining hazards and exposures; the total loss is written in the center of each circle. (a) CaMa-Flood and population-based map (PB), (b) CaMa-Flood and sectoral-based map with city effect (SBCE), (c) MODIS and population-based map (PB), (d) MODIS and sector-based map with city effect (SBCE), and (e) the World Bank report (2011).**


Firstly, comparing the losses by the different hazard data with the same exposure, the SBCE map, the service sector loss
according to CaMa-Flood (USD 15.67 billion) was over 15-fold larger than that according to MODIS (USD 0.92 billion).
This large difference was caused by the shorter average inundation period and smaller flood area in MODIS than in
CaMa-Flood. MODIS is known to tend to fail to capture the flood extent in urban areas with high densities of tall buildings





and that leads to the underestimation in inundation. In addition to different total losses, ratios of service sector loss to the
total loss differed between two results : 60.79% according to CaMa-Flood and 40.11% according to MODIS. This result
showed the sectoral ratio of the loss can be changed depending on spatially different hazards. This sectoral difference was
newly found by this study since the traditional population-based GDP map could not show this difference.

The result by the set of hazard of CaMa-Flood and exposure of the SBCE map (b in Fig. 5) was consistent with the reported
total loss, although the sectoral losses differed from the report. The total loss differed from the report by only –0.72% (USD
25.78 billion estimated loss vs. USD 25.96 billion reported loss), the service sector loss was overestimated (USD +7.57
billion loss, +29.59 point sectoral loss ratio), and the industrial sector loss was underestimated (USD –7.83 billion loss,
–29.92 point sectoral loss ratio). In the service sector, the results were overestimated for the larger inundation extent and
longer inundation period due to the lack of flood protective effect data in urban areas, where many services are located. In
the industrial sector, although the hazard in the numerical simulation captured the flood extent over the industrial sector area
and the long-lasting inundation period, the loss was underestimated. The reported value excludes assets damage but includes
economic losses other than production reduction by direct contact with the flood, such as production stoppage due to
shortages of raw materials induced by blocked roads. Therefore, if we assume that the new sectoral GDP map captured the
industrial locations and they were successfully considered to be flooded, this underestimation is presumed to be caused by a
lack of data reflecting the indirect production stoppage.

In addition to the notable omissions of urban flood protection and indirect production stoppage from the analysis, addressing
the inherent uncertainty associated with the recovery coefficient is of utmost importance. This coefficient plays a pivotal role
in calculating the recovery period following an inundation event and consequently has a substantial impact on the estimation
of business interruption losses, as demonstrated in the equation. However, determining the most appropriate coefficient
proves to be a formidable challenge, given its variability across different locations and sectors, a fact substantiated by both
Taguchi et al. (2022) and Kimura et al. (2007). Presently, attempting to ascertain the ideal coefficient for each sector is
difficult due to the absence of comprehensive observed data. It is crucial that future research investigates this matter.
**4.2 Limitation**
Firstly, there are uncertainties in the assumption of distributing sectoral GDP in proportion to the fraction of each land use. In
the Methods, we decided to consider the other components affecting the spatial accumulation such as population density only
in service GDP and assumed GDP per area is uniform in industry and agriculture. However, GDP per area could be different
depending on areas. For agriculture, it was indicated that GDP per area depends on the type of crops in the Result. Also, for
industry, produce per area was reported as different depending on sub sectors among industry. For example, in Japan,
production per area of the chemical products sector is almost five times larger than that of transport equipment (METI,





2007). These indications are difficult to utilize for the method of generating the global map because the data related to spatial
distribution of crop type and subsectors are not available, which is the different case from the service GDP map using
globally available population map. In this study, we indicated the importance of considering other components affecting
GDP per area by showing the improvement of service GDP map by City Effect and the low accuracy of agriculture GDP
map. Therefore, we expected further research on finding relationships between sectoral GDP per area and indices which
could be obtained by public and globally available data such as those provided by satellite observation or public statistics.

This study was limited in that the validation and comparison of the GDP map was performed only for Thailand and for the
map in 2010. The study methodology should be validated for other countries prior to global applications. However, this is the
first study to quantify the differences between traditionally used GDP maps and actual economic activity, and to evaluate
how such GDP maps may be improved using satellite products, for countries with large differences in sectoral GDP among
subnational districts, such as Thailand. In this point, this study could contribute to the improvement of global natural hazard
risk assessment, as the methodology and dataset used in this study can be easily applied to global. For that this study
investigated only the map in 2010, although we did not carry out the analysis on the temporal change of sectoral GDP map,
the data of land use map and national sectoral GDP we used in this study are available in other multiple years. Thus, the
method in this study is applicable also to the analysis on different time series and we expected further analysis on it in the
future.
**5 Data availability**
The global sectoral GDP maps are publicly available via Zenodo at https://doi.org/10.5281/zenodo.13991673 (Shoji et al.,
2024). The maps on Zenodo correspond to the SBCE maps in this paper and are stored as geotiff files. In total, there are nine
maps in the dataset, for each sector (service, industry, and agriculture) and year (2010, 2015, and 2020).
**6 Summary**
In this study, we generated a spatially distributed sectoral GDP map by leveraging a recently available global detailed land
use dataset; the map showed better consistency with subnational GDP statistics than the traditional GDP map did, relying
only on the gridded population map. We found that the land use classification of residential and non-residential areas could
be used to spatially distinguish the service and industrial sector areas. The accumulation of non-residential areas worked well
as a proxy of industrial sector production intensity. Conversely, that of residential areas was insufficient to express the high
accumulation of economic activity by the service sector in large cities. To overcome this problem, we considered the
city-scale effect of the intensity of service sector production. This city-scale effect expressed a realistic economic activity



accumulation in the service GDP distribution and is a globally available satellite product. For the agricultural sector, we
determined that it is necessary to incorporate crop type information.
The flood BI loss estimation using the sector-based GDP map confirmed that the new sectoral GDP map was able to express
sectoral differences in the estimated BI loss, depending on the different spatial distributions of hazard. The underestimation
of the industrial sector loss was probably resulting from a lack of data reflecting the effect of transportation network
disruption. To consider the loss due to such transportation disruption and estimate more realistic economic losses, it is
necessary to include information on both the road network and transportation of goods for the industrial sector by combining
road network data and transportation statistics between each area within each country.
This new sectoral GDP map in global can serve as a foundation for estimating economic losses classified by sector while
meticulously accounting globally for the intricacies of land use patterns. This enables precise calculations of sector-specific
losses by various natural hazards on a global scale.

**Competing interests**

The contact author has declared that none of the authors has any competing interests.

**Acknowledgement**

This work was supported by Japan Science and Technology Agency (JST) [Moonshot R&D; JPMJMS2281] and Ministry of
the Environment of Japan / Environmental Restoration and Conservation Agency [Development Fund;
JPMEERF23S21130].

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
