# Peer review of "Global spatially-distributed sectoral GDP map for disaster risk"

_Earth System Science Data, 2024_

## Author Comment (AC1)

**Reply to the comments by Referee #1**

In this document, the review comments are in black, our responses are in blue.

**Referee #1**

This paper downscales national GDP estimates across a global grid of 30 x 30 arcsec pixels. This is an interesting objective and recognizes that little work has indeed been done to move beyond people-based GDP-scaling to one that also considers the distribution of economic activities. The methodology as well as several involved assumptions and uncertainties are described transparently.

However, I have several concerns regarding this paper and the quality of the dataset. In addition to the detailed comments provided below, overall, it appears that the paper attempts to integrate two papers rather than producing a single focused paper; the paper namely both documents the creation of a global GDP map, and attributes much of the paper's attention (see e.g. the discussion section) on Thailand and Thai-specific issues.

> Thank you very much for your constructive comments which are very helpful for improving the manuscript. We made a plan for modifying the manuscript with additional background information. These are to be included in the revised version of the manuscript.

(1) The paper augments established European Commission data to differentiate global land use by residential, non-residential, and cropland uses. However, it is assumed (p.6) that residential use ("RES") represents the service sector of the economy. That is a very rough proxy given that this includes the housing of those who work in non-residential areas (the "industrial sector"), as most people do.

> We are sincerely grateful to Referee#1 for the comment. This paper focuses on "where GDP is generated" for the allocation of GDP, and does not consider "where the employees and users who generate GDP live." "Where the employees and users who generate GDP in each sector live" requires consideration of transportation networks and the estimation becomes complicated, so this is not considered in this paper.

Focusing on "where GDP is generated in each sector," it is possible to distribute simply by using existing global datasets by making the following assumptions:

•The service sector generates GDP in the Residential area where direct consumers exist.

•The industrial sector generates GDP in the Non-residential area where factories are located.

We understand that the fact that we do not consider the relationship between GDP production locations for each sector and workers' residential spaces is a limitation of this

**study, as you pointed out. We plan to add this point to the Limitations section within the Discussion part.**

Moreover, the non-residential areas being classified as the 'industrial' sector, if I understand the classification scheme correctly, pools together any services and manufacturing and other sectors as 'industrial,' separately from 'services'. This appears to be inappropriate and thus call into question whether the global map is able to distinguish between sectors. The data do appear to possibly reasonably allow for a global GDP map, without sectoral differentiation, that downscales national GDP estimates given local non-residential land use.

> Thank you for your valuable comments and suggestions. We appreciate your attention to detail and the opportunity to clarify our industry classification.

To address your concern regarding the clarity of industry classification in this paper, we have used the following definitions based on the International Standard Industrial Classification (ISIC) Rev 4 codes from the World Bank's World Development Index:

Agriculture: ISIC 01-03 (A)

Service: ISIC\* 50-99

Industry: ISIC 05-43 (B-F)

\* It should be noted that only the Service sector is based on ISIC Rev. 3

For further details, please refer to the following URL:

ISIC Rev 4, https://unstats.un.org/unsd/publication/seriesm/seriesm\_4rev4e.pdf

ISIC Rev 3,

https://unstats.un.org/unsd/classifications/Econ/Download/In%20Text/ISIC\_Rev\_3\_English.pdf

This means that our "industry" classification does not include "wholesale" or "professional services," which are categorized under "Service" in the World Bank's definitions. We believe that our dataset, with this classification method, aligns with widely used classification approaches. We acknowledge that this definition was not explicitly stated in the original manuscript. In response to your feedback, we have added the classification details mentioned above to the manuscript to ensure clarity for our readers.

Additionally, we present a comparative analysis between the National Land Use Database (NLUD) land classification data in the United States and the Global Human Settlement Layer (GHSL) data (RES/NRES categories) used in this paper.

The following table (Table R1) illustrates the proportion of areas classified as RES and NRES in GHSL within the service and industrial sectors of the NLUD land classification. The table reveals that a significant portion of service areas is identified as RES areas, while approximately half of the industrial areas are classified as NRES areas.

Table R1: Percentage (Area-Based) of GHSL Residential Area (RES) and Non-Residential Area (NRES) within NLUD Land Use Categories (Service and Industry)

|      | Service | Industry |
|------|---------|----------|
| NRES | 9.2%    | 41.6%    |
| RES  | 90.8%   | 58.4%    |

Based on this table, we argue that using RES areas as a proxy for service industries is reasonable. For NRES areas as a proxy for industrial industries, the results suggest that large-scale factories are classified as NRES areas, while small to medium-sized factories adjacent to residential areas are classified as RES areas. Therefore, assigning all industrial activities to NRES areas may not accurately represent industrial GDP in regions with small to medium-sized factories, which is a significant limitation of this study. This limitation will be added in the revised version of this paper.

However, we believe that this limitation does not significantly undermine the importance of our research. Given the absence of detailed global land classification data, finding a perfect proxy is challenging. Despite the inability to accurately represent the locations of small to medium-sized factories, which account for approximately 30% of manufacturing GDP (in the US, for example), our dataset still captures the industrial GDP distribution of large-scale factories, which contribute to the remaining 70% of GDP.

(2)Claims such as "in the United States, industrial GDP is widely dispersed regardless of urban areas" are interesting but also bold, given that the observation comes from the East coast of the USA which is relatively agglomerated (how are "cities" defined in the paper?) and paired with serious uncertainty, given that the validation of the global dataset is done for Thailand but not for the rest of the world. Ideally, analytical claims should be made only for regions for which the data are also validated to not over-assert the validity of the data that underpin the insights. In any case the validity of the findings could be asserted more carefully. It would also be helpful to compare the insights against to standing knowledge,

whether from estimates in other papers or also reports (e.g., such as the 2012 'Urban America' McKinsey report).

> Thank you for pointing out the limitation of our current validation, which focuses solely on Thailand. We acknowledge that this raises concerns about the generalizability of our findings to other regions.

To address this, we plan to expand our validation efforts in the revised manuscript. Specifically, we will conduct a similar sub-national scale validation for major regions beyond Thailand, with a primary focus on the United States, utilizing statistical data at that scale. This additional analysis will strengthen the credibility of our data and provide a more robust assessment of its applicability to diverse economic contexts.

(3) The paper could do more to underpin assumptions with field knowledge, in particular on how the assumptions could drive the outcomes observed in the global map. For instance, on p.6 it is stated that "the service GDP was distributed only in pixels within cities and the amount of distributed GDP was proportional to the population density of the city where the pixel is located". This appears to in effect assume away any service sector presence outside of urban areas, which is unrealistic, and that the amount of GDP attributed to a pixel is contingent on city density —other than the size of the city— which indeed drives productivity but not overall output levels as those instead respond predominantly to city scale.

> Thank you for your comment regarding the definition of service GDP production areas in our study. As you correctly pointed out, our current methodology confines service GDP generation to areas within urban polygons as defined by the GRUMP dataset.

We acknowledge this simplification and would like to explain our rationale. GDP generated outside of urban areas is generally significantly smaller in magnitude compared to GDP within urban centers. Furthermore, obtaining reliable proxies for distributing these minute amounts of GDP across vast rural areas presents a considerable challenge. Therefore, we believe that distributing service GDP exclusively within urban areas offers a reasonable approach when creating a spatially explicit GDP map. This assumption is supported by our comparative analysis in Thailand, where we examined the impact of including or excluding this "city effect."

Additionally, as highlighted in the manuscript (line 136), existing research indicates a strong correlation between urban population density and service sector GDP, rather than the total urban population. Furthermore, our preliminary analyses have confirmed that using population density yields service GDP classifications that more closely align with observed statistical data. When considering population density in this study, we calculated it using the population and total area of each urban polygon. Therefore, urban size is considered on a city-by-city basis (as larger cities contain larger populations and total areas). This reinforces our confidence in the chosen methodology.

**Further comments**

- The narrative flow and grammar should be checked closely throughout the manuscript (see, e.g., the first five sentences of the abstract).

> Thank you for your helpful feedback. We have revised the abstract as follows, incorporating your suggestions:

"Accurate global risk assessment of economic losses from natural disasters, accounting for diverse land uses, is crucial. However, globally consistent, high-resolution, sector-specific economic data at the pixel level remain unavailable for assessing exposure to localized hazards like floods. Here, we leverage novel land-use data to generate a global, spatially disaggregated map of sector-specific gross domestic product (GDP). We produced three 30 arcsec resolution global GDP maps for 2010, 2015, and 2020, representing the service, industry, and agriculture sectors. First, we observed that the spatial correlation between industrial GDP distribution and urban centers, where service GDP is concentrated, varies internationally. For instance, US industrial GDP is broadly dispersed independently of urban areas, while Indian industrial GDP clusters near urban areas. Second, validating against subnational statistics for Thailand, where ground-truth data exist, we found that traditional population-based GDP maps yielded a 63.0% relative error in sectoral GDP at the subnational level compared to official statistics. Our new sector-specific GDP map reduced this error to 26.2%. Subsequently, integrating this map with sector-level business interruption (BI) loss estimates from river flooding, we demonstrated that sectoral loss ratios to total loss varied considerably with flood hazard spatial distribution. Using the new GDP map, the estimated total loss approached reported values, although some discrepancies in sectoral loss ratios persist, highlighting the need for further refinement of loss-estimation models."

We are also planning to make further revisions throughout the manuscript based on your other comments.

---

## Author Comment (AC2)

**Reply to the comments by Referee #2**

In this document, the review comments are in black, our responses are in blue.

**Referee #2**

This manuscript develops a sectoral GDP map (for service, industry, and agriculture) at 30 arcsec resolution and explores its application in disaster risk analysis. The authors generate land-use data and population data to downscale national-level GDP to derive spatial distribution results. By providing high-resolution global sectoral GDP maps, this dataset offers more detailed geospatial information to support disaster risk analysis and economic loss assessments.

The methodology and limitations in the manuscript are clearly discussed. However, the validation and analysis of the data itself need to be strengthened. Additionally, the Discussion section should be reconsidered in terms of its length and content.

> Thank you very much for your constructive comments which are very helpful for improving the manuscript. We made a plan for modifying the manuscript with additional background information. These are to be included in the revised version of the manuscript.

**Specific Comments:**

1. The Introduction section should include a discussion of other existing GDP spatial datasets, covering their methodologies, spatial resolutions, and the challenges in existing GDP mapping processes.

> Thank you for your feedback. In the revised manuscript, we will include additional details about the dataset, such as its spatiotemporal resolution, and provide a more comprehensive comparison with other existing datasets.

2. The study assumes that service-sector GDP is primarily distributed in high-population-density areas, but certain economic activities—such as high-end financial services and tourism—do not necessarily follow this pattern. For example, the financial district in Manhattan has an extremely high GDP density despite relatively low residential population density. Have the authors considered such spatial distribution patterns of economic activities?

> Thank you for your insightful comment regarding the handling of service GDP. We appreciate your attention to this detail.

As you pointed out, using fine-grained municipal-level population density could indeed lead to issues. However, our approach leverages the GRUMP dataset, which defines urban polygons based on nighttime light data, effectively capturing spatially contiguous urban areas. This means that large metropolitan areas, such as the area encompassing Manhattan, are treated as a single urban entity. Therefore, while Manhattan itself may have a high concentration of service sector activity, the GRUMP polygon for this area also includes surrounding residential areas, resulting in a high overall population density for the urban entity. This, in turn, leads to a correspondingly high allocation of service sector GDP within that defined urban area. We believe this approach provides a reasonable representation of the spatial distribution of service sector GDP at the scale of analysis used in this study.

3. Why did the authors choose the GRUMP dataset to account for urban effects instead of other datasets? A brief explanation for this choice would strengthen the methodology section.

> Thank you for your question regarding the choice of urban polygon dataset. We considered several options, including:

- 1. GRUMP
- 2. GHS-Urban
- 3. World Urban Areas (available in Esri ArcGIS)

We ultimately selected GRUMP for the following reasons. The GHS-Urban dataset, while comprehensive, delineates urban areas at a very fine-grained level. This resulted in the splitting of what are generally considered single urban agglomerations into multiple, separate urban polygons. This fragmentation led to unrealistically high population densities in some polygons when implementing the city-effect, which in turn skewed our service GDP estimates. Therefore, we deemed GHS-Urban unsuitable for our specific application.

The World Urban Areas dataset offered polygons that were very similar to those in GRUMP. However, as it is not openly accessible, we opted for the open-source GRUMP dataset to maintain transparency and reproducibility in our research.

4. The validation was conducted in only seven regions of Thailand, but Thailand's economic structure may not be representative at a global scale. For example, Western economies are more dependent on the service sector, while industrial and agricultural distributions vary significantly across different regions. Have the authors considered additional validation in countries with different economic structures, such as the United States, China, or Germany?

> We appreciate your observation regarding the limited scope of our current validation, which is confined to Thailand. We recognize that this raises questions about the broader applicability of our results. In the revised manuscript, we will address this by extending our validation to include other key regions, notably the United States. This expanded analysis, using sub-national statistical data, will bolster the reliability of our dataset and offer a more comprehensive evaluation of its performance across diverse economic settings.

5. A comparison with other existing GDP products or remote sensing proxies should be included to better highlight this dataset's advantages.

> Thank you for your comment regarding the comparison with existing GDP products and remote sensing proxies. We understand your question and would like to clarify our approach.

As you mentioned, GDP distribution has traditionally been conducted at scales ranging from national to municipal levels, based on statistical information. Studies that generate GDP maps at the grid scale, as we do in this paper, are limited to those mentioned in the Introduction.

Regarding remote sensing proxies, existing research generally falls into two categories: land cover or population distribution. Previous studies have primarily focused on population distribution. Our work represents, to the best of our knowledge, the first attempt to utilize land cover as a primary proxy for generating a global, high-resolution GDP map.

Therefore, when comparing our work to existing GDP products and remote sensing proxies, the most relevant comparison is indeed the one we already provide in the manuscript with our population-based map. This comparison serves to highlight the key differences and potential advantages of using land cover as a proxy, as opposed to the more traditional approach based on population distribution. We believe this comparison effectively addresses the spirit of your question regarding comparison with existing products and proxies.

6. Since the study aims to provide a globally applicable dataset, the Thailand case study in Section 4.1 should be presented as a supporting example rather than the main focus. It is recommended that the authors strengthen the discussion of the dataset itself, particularly regarding accuracy assessment, comparisons with existing datasets, spatial details, and temporal variation analysis. Additionally, by reducing the emphasis on the Thailand case study and discussing broader disaster analysis applications, the authors can better highlight the dataset's global applicability.

> Thank you for your helpful feedback. As mentioned in our response to another comment, we are planning to add validation for regions beyond Thailand in the revised manuscript. To accommodate this and maintain a balanced focus, we will reduce or remove the content related to the flood damage analysis in Thailand. This will allow us to shift the emphasis of the manuscript towards the broader validation efforts, including the comparison with population-based maps, and provide a more comprehensive assessment of the dataset's global applicability.

---

## Author Response (AR1)

In this document, the review comments are in black, our responses are in blue. The changes made in the manuscript for this revision are written in red.

**Reply to the comments by Referee #1**

This paper downscales national GDP estimates across a global grid of 30 x 30 arcsec pixels. This is an interesting objective and recognizes that little work has indeed been done to move beyond people-based GDP-scaling to one that also considers the distribution of economic activities. The methodology as well as several involved assumptions and uncertainties are described transparently.

However, I have several concerns regarding this paper and the quality of the dataset. In addition to the detailed comments provided below, overall, it appears that the paper attempts to integrate two papers rather than producing a single focused paper; the paper namely both documents the creation of a global GDP map, and attributes much of the paper's attention (see e.g. the discussion section) on Thailand and Thai-specific issues.

> Thank you very much for your constructive comments which are very helpful for improving the manuscript.

We've made significant revisions since the last version, incorporating various comments. The two main changes are:

- 1. Modified spatial distribution methods for Service and Agricultural GDP.
- 2. Expanded validation scope from only Thailand to a global scale.

These changes have refined the rationale behind our spatial distribution methods for each sector's GDP. As a result, we've confirmed that the spatial distributions now align with sub-national statistical data across numerous regions, not just Thailand. The following sections will detail how the new manuscript addresses specific concerns raised in previous comments. For more details of the validation, please refer to the Results section in the main text.

- ①The paper augments established European Commission data to differentiate global land use by residential, non-residential, and cropland uses. However, it is assumed (p.6) that residential use ("RES") represents the service sector of the economy. That is a very rough proxy given that this includes the housing of those who work in non-residential areas (the "industrial sector"), as most people do.
- > We are sincerely grateful to you for the comment. This paper focuses on "where GDP is generated" for the allocation of GDP, and does not consider "where the employees and users who generate GDP live." "Where the employees and users who generate GDP in each sector live" requires consideration of transportation networks and the estimation becomes complicated, so this is not considered in this paper.

Focusing on "where GDP is generated in each sector," it is possible to distribute simply by using existing global datasets by making the following assumptions:

- •The service sector generates GDP in the Residential area where direct consumers exist.
- The industrial sector generates GDP in the Non-residential area where factories are located.

We understand that the fact that we do not consider the relationship between GDP production locations for each sector and workers' residential spaces is a limitation of this study, as you pointed out. We added this point to the Discussion part, as shown below.

"Related to this limitation of the indirect production stoppage, it is important to recognize that the methodology, including that of this paper and previous studies, which determines the GDP produced in each pixel using indicators such as GDP per unit area, overlooks the fact that labor supplied from remote locations is necessary for GDP production. To rephrase this with the example of a factory affected by a disaster: while the GDP output itself occurs at the factory's location, the workers who carry out the production reside in surrounding or remote areas. Therefore, if a disaster occurs in these remote residential areas, the GDP output should cease. However, pixel-based calculation methods would fail to represent this cessation of GDP output as long as the factory's pixel is unaffected. This is considered a non-negligible impact in regions where economic activity and residential areas are clearly separated, but quantifying this impact on a global scale is currently challenging. Alongside future research on regional differences in GDP per unit area, this remains a limitation that we must consider moving forward."

Moreover, the non-residential areas being classified as the 'industrial' sector, if I understand the classification scheme correctly, pools together any services and manufacturing and other sectors as 'industrial,' separately from 'services'. This appears to be inappropriate and thus call into question whether the global map is able to distinguish between sectors. The data do appear to possibly reasonably allow for a global GDP map, without sectoral differentiation, that downscales national GDP estimates given local non-residential land use.

> Thank you for your valuable comments and suggestions. We appreciate your attention to detail and the opportunity to clarify our industry classification.

To address your concern regarding the clarity of industry classification in this paper, we have used the following definitions based on the International Standard Industrial Classification (ISIC) Rev 4 codes from the World Bank's World Development Index:

Agriculture: ISIC 01-03 (A)

Service: ISIC\* 50-99

Industry: ISIC 05-43 (B-F)

For further details, please refer to the following URL:

ISIC Rev 4, https://unstats.un.org/unsd/publication/seriesm/seriesm\_4rev4e.pdf ISIC Rev 3,

https://unstats.un.org/unsd/classifications/Econ/Download/In%20Text/ISIC\_Rev\_3\_English.pdf

This means that our "industry" classification does not include "wholesale" or "professional services," which are categorized under "Service" in the World Bank's definitions. We believe that our dataset, with this classification method, aligns with widely used classification approaches. We acknowledge that this definition was not explicitly stated in the original manuscript. In response to your feedback, we have added the classification details mentioned above to the manuscript to ensure clarity for our readers, as shown below.

| Sector      | Definition of ISIC |  |
|-------------|--------------------|--|
| Agriculture | ISIC 01-03 (A)     |  |
| Service     | ISIC* 50-99        |  |
| Industry    | ISIC 05-43 (B-F)   |  |

\*It should be noted that only the Service sector is based on ISIC Rev. 3.

Table 2: Definition of each sector, based on the International Standard Industrial Classification (ISIC) Rev 4, in the GDP data by the World Bank (2023).

Additionally, we present a comparative analysis between the National Land Use Database (NLUD) land classification data in the United States and the Global Human Settlement Layer (GHSL) data (RES/NRES categories) used in this paper.

The following table (Table R1) illustrates the proportion of areas classified as RES and NRES in GHSL within the service and industrial sectors of the NLUD land classification. The table reveals that a significant portion of service areas is identified as RES areas, while approximately half of the industrial areas are classified as NRES areas.

Table R1: Percentage (Area-Based) of GHSL Residential Area (RES) and Non-Residential Area (NRES) within NLUD Land Use Categories (Service and Industry)

\* It should be noted that only the Service sector is based on ISIC Rev. 3

|      | Service | Industry |
|------|---------|----------|
| NRES | 9.2%    | 41.6%    |
| RES  | 90.8%   | 58.4%    |

Based on this table, we argue that using RES areas as a proxy for service industries is reasonable. For NRES areas as a proxy for industrial industries, the results suggest that large-scale factories are classified as NRES areas, while small to medium-sized factories adjacent to residential areas are classified as RES areas. Therefore, assigning all industrial activities to NRES areas may not accurately represent industrial GDP in regions with small to medium-sized factories, which is a significant limitation of this study. This limitation has been added in the revised version of this paper.

However, we believe that this limitation does not significantly undermine the importance of our research. Given the absence of detailed global land classification data, finding a perfect proxy is challenging. Despite the inability to accurately represent the locations of small to medium-sized factories, which account for approximately 30% of manufacturing GDP (in the US, for example), our dataset still captures the industrial GDP distribution of large-scale factories, which contribute to the remaining 70% of GDP.

②Claims such as "in the United States, industrial GDP is widely dispersed regardless of urban areas" are interesting but also bold, given that the observation comes from the East coast of the USA which is relatively agglomerated (how are "cities" defined in the paper?) and paired with serious uncertainty, given that the validation of the global dataset is done for Thailand but not for the rest of the world. Ideally, analytical claims should be made only for regions for which the data are also validated to not over-assert the validity of the data that underpin the insights. In any case the validity of the findings could be asserted more carefully. It would also be helpful to compare the insights against to standing knowledge, whether from estimates in other papers or also reports (e.g., such as the 2012 'Urban America' McKinsey report).

> Thank you for pointing out the limitation of our current validation, which focuses solely on Thailand. We acknowledge that this raises concerns about the generalizability of our findings to other regions.

To address this, we expanded our validation globally. This revealed that the map exhibits a distribution consistent with actual sub-national statistics across many regions outside of Thailand. Consequently, we removed the specific reference to the U.S. and instead focused on Thailand, Japan, and the agricultural sector in Paris, where the map's accuracy is well-established, shown as follows. Furthermore, these specific mentions are confirmed to align with the generally recognized characteristics of each respective region.

"In the figure of Japan, Japan's three major metropolitan areas—Tokyo, Osaka, and Aichi—shows variations in sectoral distribution, despite their common characteristic of high population concentration. In the GDP map, the service sector predominates in the coastal areas of Tokyo and Osaka, which are marked by high population and service industry presence. In contrast, Aichi's coastal regions exhibit a widespread predominance of industrial GDP. Industrial GDP is not uniformly distributed across the entire Aichi area. Within Aichi, the more inland urban center, such as the Nagoya area, shows a prevalence of the service sector, with industrial GDP concentrated in coastal areas. These findings align with Aichi's higher proportion of industrial GDP compared to Tokyo and Osaka (DOSE, 2024), and the formation of an extensive industrial belt along its coastal regions. This dataset facilitates the depiction of detailed distributional differences within these areas.

When comparing central Bangkok with its southeastern region, a similar pattern emerges as a case in Japan. The southeastern area, specifically the Eastern Seaboard and Eastern Economic Corridor (EEC) centered around Laem Chabang Port, has developed as an industrial hub. In this region, industrial GDP predominates over service sector GDP. Regarding the distribution of agricultural GDP, Japan shows fewer pixels where agricultural GDP is dominant, largely because much of its agricultural land is located relatively close to urban areas. However, in Thailand and France, extensive areas with dominant agricultural GDP are observed around metropolitan centers like Bangkok and Paris. For instance, Figure 4, which shows only agricultural GDP for France, illustrates that agricultural GDP is minimally developed around densely populated Paris. Conversely, it depicts widespread agricultural activity in the less populated surrounding regions."

- ③The paper could do more to underpin assumptions with field knowledge, in particular on how the assumptions could drive the outcomes observed in the global map. For instance, on p.6 it is stated that "the service GDP was distributed only in pixels within cities and the amount of distributed GDP was proportional to the population density of the city where the pixel is located". This appears to in effect assume away any service sector presence outside of urban areas, which is unrealistic, and that the amount of GDP attributed to a pixel is contingent on city density —other than the size of the city— which indeed drives productivity but not overall output levels as those instead respond predominantly to city scale.
- > Thank you for your comment regarding the definition of service GDP production areas in our study. As you correctly pointed out, our previous methodology confined service GDP generation to areas within urban polygons as defined by the GRUMP dataset. However, through the process of re-validation and improvements in the GDP distribution methodology, we changed the methods and modified thoroughly the rationale of the distribution methods for agriculture and service sectors, including, as follows.

**"2.1.3 Land-use-based agriculture sector GDP**

To better reflect the spatial structure of production activities, we introduce the supplier effect, which assumes a beneficiary-supplier relationship. Specifically, agricultural production occurring in peri-urban or rural areas surrounding major population centers is regarded as supplying food and resources to those urban beneficiaries. These agricultural zones, while themselves sparsely populated, are functionally integrated with the urban economy. Therefore, they are expected to exhibit higher GDP values than similarly sparse regions that are not spatially or economically connected to urban demand. To capture this spatial interdependence, the supplier effect applies a distance-decay reallocation from beneficiary pixels (population-based GDP map) to nearby supply-side pixels, namely those identified as MCROP. Technically, this is implemented as a linear decay function, in which full weight is given within an inner threshold of 150 km, and weight decrease linearly to zero at an outer threshold of 300km.

$$w_{ij} = if d_{ij} \le d_{in}: 1; if d_{in} < d_{ij} \le d_{out}: 1 - (d_{ij} - d_{in}) / d_{in}; if d_{ij} > d_{out}: 0$$
 (2)

**2.1.4 Land-use-based service sector GDP**

Similarly, PB of the service sector is reallocated to residential areas (RES) by applying the supplier effect. The rationale here differs slightly from that for agriculture. Grid-scale population data (e.g., at 30-arcsecond resolution, or approximately 1 × 1 km per pixel) are too fine to represent realistic service usage, since people commonly travel more than 1 km by car or public transportation to access services (Ciccone and Hall, 1996). Therefore, this reallocation is designed to represent commuting patterns, where service activities in peri-urban zones support nearby urban demand centers. In this context, we use a supplier effect with an inner threshold of 25 km (representing high-intensity interaction) and an outer threshold of 50 km, beyond which service contributions are assumed negligible."

**Further comments**

- The narrative flow and grammar should be checked closely throughout the manuscript (see, e.g., the first five sentences of the abstract).
- > Thank you for your helpful feedback. We have revised the abstract as follows, incorporating your suggestions:

"Global risk assessments of economic losses by natural disasters while considering various land uses is essential. However, sector-specific, high-resolution pixel-level economic data are not yet available globally to assess exposure to local disasters such as floods. In this study, we employed new land-use data to construct global, spatially distributed map of sector-specific gross domestic product (GDP). We developed three global GDP maps, SectGDP30, in 2010, 2015, and 2020 for service, industry, and agriculture sector, with 30

arcsec resolution. The map (SectGDP30) demonstrates strong consistency (R^2 > 0.9) with actual sub-national statistical data, exhibiting superior alignment compared to conventional GDP maps (PB-method) reliant solely on gridded population information. The methodology refined GDP distribution for specific sectors. Industry GDP was more accurately mapped using non-residential land areas as a proxy, effectively capturing its localized concentrations. Agriculture GDP's accuracy improved by incorporating cropland data and a distance-based distribution assumption from population agglomeration. Application of this dataset in estimating flood-induced business interruption (BI) losses confirmed the map's capacity to represent inter-sectoral differences in estimated losses, reflecting varied hazard spatial distributions. This underscores the importance of considering sector-specific spatial patterns for accurate disaster damage assessment. These maps serve as a foundational tool for estimating detailed, sector-classified economic losses, enabling precise calculation of sector-specific impacts from diverse natural disasters worldwide. These global sectoral GDP maps (SectGDP30) are available at https://doi.org/10.5281/zenodo.13991673 (Shoji et al., 2024)."

We also made some modifications throughout the manuscript based on your other comments.

**Reply to the comments by Referee #2**

This manuscript develops a sectoral GDP map (for service, industry, and agriculture) at 30 arcsec resolution and explores its application in disaster risk analysis. The authors generate land-use data and population data to downscale national-level GDP to derive spatial distribution results. By providing high-resolution global sectoral GDP maps, this dataset offers more detailed geospatial information to support disaster risk analysis and economic loss assessments.

The methodology and limitations in the manuscript are clearly discussed. However, the validation and analysis of the data itself need to be strengthened. Additionally, the Discussion section should be reconsidered in terms of its length and content.

> Thank you very much for your constructive comments which are very helpful for improving the manuscript.

We've made significant revisions since the last version, incorporating various comments. The two main changes are:

- 1. Modified spatial distribution methods for Service and Agricultural GDP.
- 2. Expanded validation scope from only Thailand to a global scale.

These changes have refined the rationale behind our spatial distribution methods for each sector's GDP. As a result, we've confirmed that the spatial distributions now align with sub-national statistical data across numerous regions, not just Thailand. The following sections will detail how the new manuscript addresses specific concerns raised in previous comments.

**Specific Comments:**

- 1. The Introduction section should include a discussion of other existing GDP spatial datasets, covering their methodologies, spatial resolutions, and the challenges in existing GDP mapping processes.
- > Thank you for your feedback. We've updated the description of GDP dataset products in Introduction, based on your comments, as follows.

"GDP maps developed using these methods are generally created for specific purposes, such as disaster damage estimation, and are therefore not typically released as standalone datasets or products. Among those that are publicly available, "Downscaled gridded global dataset for gross domestic product (GDP) per capita PPP over 1990–2022" by Kummu et al., 2025, is notable. This dataset generates gridded GDP map products with resolutions ranging from 30 arcmin to 30 arcsec for each year since 1990, based on sub-national statistics released by various countries and utilizing population count maps."

Previously, existing research primarily created GDP maps using population data as a proxy. These maps were generally developed as simplified tools for specific purposes, such as disaster damage estimation, and were not typically released as public datasets or products. Consequently, methods relying solely on population data as a proxy were widely adopted. Currently, the only publicly available pixel-level global GDP distribution map product is Kummu et al., 2025 (previously Kummu et al., 2018). We believe the limitations of this product and the methods using population data as a proxy are already addressed in the original manuscript.

- 2. The study assumes that service-sector GDP is primarily distributed in high-population-density areas, but certain economic activities—such as high-end financial services and tourism—do not necessarily follow this pattern. For example, the financial district in Manhattan has an extremely high GDP density despite relatively low residential population density. Have the authors considered such spatial distribution patterns of economic activities?
- > Thank you for your insightful comment regarding the handling of service GDP. We appreciate your attention to this detail.

As you pointed out, using fine-grained municipal-level population density could indeed lead to issues. However, our approach leverages the GRUMP dataset, which defines urban

polygons based on nighttime light data, effectively capturing spatially contiguous urban areas. This means that large metropolitan areas, such as the area encompassing Manhattan, are treated as a single urban entity. Therefore, while Manhattan itself may have a high concentration of service sector activity, the GRUMP polygon for this area also includes surrounding residential areas, resulting in a high overall population density for the urban entity. This, in turn, leads to a correspondingly high allocation of service sector GDP within that defined urban area. We believe this approach provides a reasonable representation of the spatial distribution of service sector GDP at the scale of analysis used in this study.

**June 30, 2025 Addendum**: While the spatial distribution method for service sector GDP, as referenced in this comment, has been revised in the current update, the fundamental approach of distributing it proportionally to the population within a given area remains unchanged. Therefore, the explanation above still applies to the new manuscript.

- 3. Why did the authors choose the GRUMP dataset to account for urban effects instead of other datasets? A brief explanation for this choice would strengthen the methodology section.
- > Thank you for your question regarding the choice of urban polygon dataset. We considered several options, including:
  - 1. GRUMP
  - 2. GHS-Urban
  - 3. World Urban Areas (available in Esri ArcGIS)

We ultimately selected GRUMP for the following reasons. The GHS-Urban dataset, while comprehensive, delineates urban areas at a very fine-grained level. This resulted in the splitting of what are generally considered single urban agglomerations into multiple, separate urban polygons. This fragmentation led to unrealistically high population densities in some polygons when implementing the city-effect, which in turn skewed our service GDP estimates. Therefore, we deemed GHS-Urban unsuitable for our specific application.

The World Urban Areas dataset offered polygons that were very similar to those in GRUMP. However, as it is not openly accessible, we opted for the open-source GRUMP dataset to maintain transparency and reproducibility in our research.

**June 30, 2025 Addendum**: As a result of this revision, the GRUMP dataset is no longer used for the spatial distribution of service sector GDP.

4. The validation was conducted in only seven regions of Thailand, but Thailand's economic structure may not be representative at a global scale. For example, Western economies are

more dependent on the service sector, while industrial and agricultural distributions vary significantly across different regions. Have the authors considered additional validation in countries with different economic structures, such as the United States, China, or Germany?

- > We appreciate your observation regarding the limited scope of our current validation, which is confined to Thailand. We recognize that this raises questions about the broader applicability of our results. Considering your valuable comment, we expand the target area of validation from only Thailand to worldwide. The result showed the strong consistency with the sub-national scale statistics in many areas in the world. For more details, please refer to the Results section in the main text.
- 5. A comparison with other existing GDP products or remote sensing proxies should be included to better highlight this dataset's advantages.
- > Thank you for your comment regarding the comparison with existing GDP products and remote sensing proxies. We understand your question and would like to clarify our approach.

As you mentioned, GDP distribution has traditionally been conducted at scales ranging from national to municipal levels, based on statistical information. Studies that generate GDP maps at the grid scale, as we do in this paper, are limited to those mentioned in the Introduction.

Regarding remote sensing proxies, existing research generally falls into two categories: land cover or population distribution. Previous studies have primarily focused on population distribution. Our work represents, to the best of our knowledge, the first attempt to utilize land cover as a primary proxy for generating a global, high-resolution GDP map.

Therefore, when comparing our work to existing GDP products and remote sensing proxies, the most relevant comparison is indeed the one we already provide in the manuscript with our population-based map. This comparison serves to highlight the key differences and potential advantages of using land cover as a proxy, as opposed to the more traditional approach based on population distribution. We believe this comparison effectively addresses the spirit of your question regarding comparison with existing products and proxies.

6. Since the study aims to provide a globally applicable dataset, the Thailand case study in Section 4.1 should be presented as a supporting example rather than the main focus. It is recommended that the authors strengthen the discussion of the dataset itself, particularly regarding accuracy assessment, comparisons with existing datasets, spatial details, and temporal variation analysis. Additionally, by reducing the emphasis on the Thailand case

study and discussing broader disaster analysis applications, the authors can better highlight the dataset's global applicability.

> Thank you for your helpful feedback. As mentioned in our response to another comment, we added validation for regions beyond Thailand in the revised manuscript. To accommodate this and maintain a balanced focus, we reduced the content related to the flood damage analysis in Thailand. This allowed us to shift the emphasis of the manuscript towards the broader validation efforts, including the comparison with population-based maps, and provide a more comprehensive assessment of the dataset's global applicability.

---

## Author Response (AR2)

- 1 The revisions made for this finalization are outlined below. The corrections include the addition of the section on
- 2 Contribution that was pointed out, a change to the contact information, and the revision of several citations (due to
- 3 discrepancies between the in-text notation and the Reference list entry).

I affirm that no content changes were made in this revision.

**7 Global spatially-distributed sectoral GDP map for disaster risk**

**8 analysis**

- 9 Takeshi Shoji1,2, Kiyoharu Kajiyama2, Dai Yamazaki1,2, Yuki Kita2,3, Megumi Watanabe2,4
- 10 1Graduate School of Engineering, The University of Tokyo, Tokyo, 113-8656, Japan
- 11 2Institute of Industrial Science, The University of Tokyo, Tokyo, 153-8505, Japan
- 12 3Gaia Vision Inc., Tokyo, Japan
- 13 4LERMA, Observatoire de Paris, Paris, 75014, France
- 14 Correspondence to: Takeshi Shoji (oi.oh.take@gmail.comtshoji@rainbow.iis.u-tokyo.ac.jp)
- 15 Abstract. Global risk assessments of economic losses by natural disasters while considering various land uses is essential.
- 16 However, sector-specific, high-resolution pixel-level economic data are not yet available globally to assess exposure to local
- 17 disasters such as floods. In this study, we employed new land-use data to construct global, spatially distributed map of
- 18 sector-specific gross domestic product (GDP). We developed three global GDP maps, SectGDP30, in 2010, 2015, and 2020
- 19 for service, industry, and agriculture sector with 30 arcsec resolution. •The map (SectGDP30) demonstrates strong
- 20 consistency ( $R^2 > 0.9$ ) with actual sub-national statistical data, exhibiting superior alignment compared to conventional
- 21 GDP maps (PB-method) reliant solely on gridded population information. The methodology refined GDP distribution for
- 22 specific sectors. Industry GDP was more accurately mapped using non-residential land areas as a proxy, effectively capturing
- 23 its localized concentrations. Agriculture GDP's accuracy improved by incorporating cropland data and a distance-based
- 24 distribution assumption from population agglomeration. Application of this dataset in estimating flood-induced business
- 25 interruption (BI) losses confirmed the map's capacity to represent inter-sectoral differences in estimated losses, reflecting
- 26 varied hazard spatial distributions. This underscores the importance of considering sector-specific spatial patterns for
- 27 accurate disaster damage assessment. These maps serve as a foundational tool for estimating detailed, sector-classified economic losses, enabling precise calculation of sector-specific impacts from diverse natural disasters worldwide. These 29 global sectoral GDP maps (SectGDP30) are available at https://doi.org/10.5281/zenodo.15774017 (Shoij et al., 2025).

**30 1 Introduction**

In recent years, as natural disasters have become more frequent and found throughout the world (IPCC, 2012), global spatial 32 data including land use and socioeconomic information have become essential for estimating the extent of disaster damage 33 and losses. With the increasing frequency and impact of localized natural disasters such as floods, high-resolution data 34 capturing the spatial distribution of socioeconomic factors are essential. However, socioeconomic data published by 35 international organizations such as the World Bank are often available only at the national or large municipal level. At the 36 research level, economic data at the municipal level have been studied (Wenz et al., 2023); however, obtaining grid-level 37 data at a resolution of several kilometers has been still challenging.

For example, as for the impact-assessment of flood disasters, researchers have undertaken a series of studies by spatially calculating the amount of asset quantity and production activity overlapped with inundated areas, leveraging global maps. Achieving this necessitates the downscaling of national-level data of economic activity, mainly gross domestic product (GDP), to finer subnational or grid-based levels. This type of product by downscaling GDP is called a "spatially distributed GDP map". This downscaling practice typically relies on gridded population data (Tanoue et al., 2021; Willner et al., 2018). Alternatively, it has involved the assembly and interpolation of available subnational statistics (Duan et al., 2022; Kummu et al., 2018) or the assumption that average building heights correlate with economic activity intensity (Taguchi et al., 2022). GDP maps developed using these methods are generally created for specific purposes, such as disaster damage estimation, and are therefore not typically released as standalone datasets or products. Among those that are publicly available, "Downscaled gridded global dataset for gross domestic product (GDP) per capita PPP over 1990–2022" by Kummu et al. (2025), is notable. This dataset generates gridded GDP map products with resolutions ranging from 30 arcmin to 30 arcsec for each year since 1990, based on sub-national statistics released by various countries and utilizing population count maps.

While these studies estimated the total amount of economic losses without considering the difference between sectors, the 53 sector-classified economic losses also need to be estimated because indirect economic losses, such as global supply chain 54 impact caused by the stoppage of production activity (Willner et al., 2018), can vary significantly depending upon the sector 55 directly affected by the flood (Sieg et al., 2019). However, spatial data of sectors by downscaling national-level data have 56 been lacking. Consequently, in the context of global studies, the estimation of sector-specific losses was achieved by 57 extrapolating the values of sectoral occupation fractions within urban area grids, as reported in the European Union, to other 58 regions (Alfieri et al., 20176; Dottori et al., 2018). Alternatively, it is assumed that specific groups of sectors experience uniform damage ratios (Willner et al., 2018; Tanoue et al., 2020). These methods did not consider the different spatial 60 accumulation between each sector and each region, which could lead to the misestimation of sector-classified losses 61 (Jongman et al., 2012; Willner et al., 2018).

The dearth of global spatial data of the economic sector arises from the absence of worldwide maps with comprehensive land use categorizations (Wenz and Willner, 2022). While regional maps provide sectoral land use classifications, including commercial and industrial areas within urban regions (e.g., The European Environmental Agency, 2017; Theobald, 2014; De Moel H et al., 2014; Ministry of Land, Infrastructure, Transport and Tourism Herr, 2021), these classifications are conspicuously absent from global maps (e.g., Bontemps et al., 2011; Esch et al., 2017). Here we focused on the recent emergence of a global land use map featuring detailed urban area classifications (Pesaresi and Politis, 2022). This development is made possible by the application of machine learning techniques that extrapolate relationships between satellite observations and actual land uses, a methodology initially established by the data in the European Union and the United States (The European Environmental Agency, 2017; Theobald, 2014) and subsequently extended to a global scale. Although this dataset facilitates a comprehensive consideration of detailed land-use patterns within urban areas worldwide, no study has yet integrated this dataset with socioeconomic data. Such integration holds the potential to pioneer a novel approach to estimating natural disaster damage accurately with sectoral classifications.

The objective of this study is to leverage a recently available global detailed land use map dataset to construct a spatially 77 distributed sectoral GDP map (SectGDP30). The accuracy of the GDP mapping of SectGDP30 is evaluated using global 78 sub-national scale statistics from DOSE dataset (Wenz et al., 2023). Furthermore, to discuss the applicability of SectGDP30 79 for practical economic loss estimation, this study examines the estimation of business interruption losses incurred due to a 80 flood event in Thailand and compares these estimations with reported values.

**81 2 Methods**

**82 2.1 Spatially distributed sectoral GDP map**

The spatially distributed sectoral GDP map was created in two steps (Figure 1). First, we classified country level GDP data 84 into three sectors: the agriculture, service, and industry sector, and they are downscaled to a spatial resolution of 30 arcsec 85 based on population data, referred as population-based map (PB-method). Second, downscaled estimates are reallocated to 86 the corresponding land use fraction maps derived from satellite products, referred to as land-use-based map (LUB-method). 87 For both the agriculture and service sectors, we generated PB-method and subsequently reallocated them using land-use data. 88 This two-step allocation is necessary because GDP is generally correlated with population distribution (Chen et al., 2022; 89 Kummu et al., 2025), and service-sector GDP, in particular, is strongly influenced by urban agglomeration effects (Morikawa, 2011). However, previous studies have shown that at high spatial resolutions, population data alone may not 91 adequately preserve these correlations (Murakami and Yamagata, 2019; Ru et al., 2022). Therefore, integrating land-use 92 information is essential to ensure spatial consistency. Unlike the agriculture and service sectors, industry sector GDP doesn't 93 necessarily follow population distribution. It often expands into suburban or rural areas with low population density (Zhuang 94 and Ye, 2023). Accordingly, we bypass the PB-method step and directly allocate country-level industrial GDP to land use 95 data. The List of the datasets used in this method is shown in Table 1.

Figure 1: Flowchart of (top) data processing and (bottom) creation of spatial distributed gross domestic product (GDP) maps of 100 Thailand for the (a) service, (b) industrial, and (c) agricultural sectors.

| Data                                                 | Format                  | Datatype | Values range                                                     | Spatial resolution | Temporal resolution                | Data source,
Reference                                           |
|------------------------------------------------------|-------------------------|----------|------------------------------------------------------------------|--------------------|------------------------------------|---------------------------------------------------------------------|
| Built up
surface area                             | Raster                  | UInt16   | 0-1000                                                           | 100m               | Five years interval (1975-2020)    | Global Human
Settlement Layer
(Pesaresi and Politis,
2022) |
| Non-residentia
surface area                     | Raster                  | UInt16   | 0-1000                                                           | 100m               | Five years interval (1975-2020)    | Global Human
Settlement Layer
(Pesaresi and Politis,
2022) |
| Crop land area                                       | Raster                  | Boolean  | 0,1
(0 - no
cropland,
1- cropland)                      | 0.9 arcsec         | Four years interval (2003-2019)    | Potapov et al., 2022                                                |
| Population count                                     | Raster                  | Float64  | 0-Inf                                                            | 30 arcsec          | Five years interval (1975-2020)    | Global Human
Settlement Layer
(Pesaresi and Politis,
2022) |
| Administrative units                                 | Vector
(Polygo
n) | -        | -                                                                | -                  | -                                  | GADM 4.1 (2023)
Level 1 Layer                                    |
| Data                                                 | Format                  | Datatype | Values range                                                     | Spatial resolution | Temporal resolution                | Data source, Reference                                              |
| Built up surface area  Non-residential surface  area | Raster                  | UInt16   | 0-1000                                                           | 100m               | five years interval
(1975-2020) | Global Human Settlement Layer
(Pesaresi and Politis, 2022)       |
| Crop land area                                       | Raster                  | Boolean  | 0,1
(0 - no cropland e
- croplan e | 0.9 arcsec         | five years interval (2003-2019)    | Potapov et al., 2022                                                |
| Population count                                     | Raster                  | Float64  | 0-Inf                                                            | 50arcsec           | five years interval
(1975-2020) | Global Human Settlement Layer
(Pesaresi and Politis, 2022)       |
| Administrative units                                 | Vector (Polygon)        | -        | -                                                                | -                  | -                                  | GADM 4.1 (2023)
Level 1 Layer                                    |

Table 1: List of the datasets used in this study.

**106 2.1.1 Population-based sectoral GDP**

In the first step, country-level GDP was partitioned into three sectors and then spatially distributed in proportion to population 108 data at a spatial resolution of 30 arcsec. We used GDP data published by the World Bank (2023), which includes both annual GDP 109 values and their sectoral ratios for the service, industrial, and agricultural sectors, and the Global Human Settlement Layer (GHSL) population grid (R2023; Pesaresi and Politis, 2022) as the source of the global gridded population map. The definition of each sector is shown in Table 2. This downscaling method has been widely employed in previous studies (Kummu et al., 2018;

Murakami and Yamagata, 2019) and will be utilized in a later section for comparison with the new method proposed in this study.

**113 2.1.2 Sectoral land use fraction map**

In the second, step, we reallocated PB-method to global sectoral land use fraction map. We generated a sectoral land use fraction map classified into three sectors (service, industry, and agriculture) and three land use type maps with different spatial resolutions:

residential (RES), non-residential (NRES), and cropland (CROP). To distinguish RES and NRES areas, we used Global Human

Settlement Layer (GHSL) (Pesaresi and Politis, 2022) built-up surface (R2022) data. This layer has 100 × 100 m resolution; each pixel has a value of 0-10,000 m2 and residential or non-residential areas may be present within one pixel. For the CROP area, we used the global map of cropland extent (Potapov et al., 2022), provided by Global Land Analysis & Discovery, which has a global spatial resolution of 0.9 arcsec. Maps with the three classes were resampled and combined into a single global sectoral land use (residential, non-residential, and cropland) fraction map at 30 arcsec resolution.

First, we upscaled the land use maps and simultaneously converted the value of each pixel in both maps into the sectoral fraction within one pixel. In each pixel, RES and NRES had values of 0-10000 m2 and CROP had a value of 0 or 1 (not cropland or cropland). We upscaled the land use maps to 30 arcsec resolution from RES and NRES at a resolution of 100 × 100 m and CROP

at a resolution of 0.9 arcsec using the GDAL averaging method (GDAL/OGR contributors, 2024). Using the 30 arcsec maps, we calculated the area attributed to each land use type in one pixel with a size of 1 × 1 arcsec and obtained land use fractions for each pixel. Because RES/NRES and CROP had different data sources, the total of the three land use type fractions was greater than one in some pixels. Therefore, we assumed that the CROP fraction could fill only areas that were not designated as RES or NRES.

Under this assumption, we modified the CROP fraction in each pixel as follows:

$$MCROP_i = min(CROP_i, (1 - RES_i - NRES_i))$$
 (1)

where MCROP, is the modified CROP fraction in pixel i, CROP, is the original CROP fraction, RES, is the RES fraction, and

$NRES_i$  is the NRES fraction.

After this modification, RES, NRES, and MCROP were considered to represent the service, industrial, and agricultural land use sectors, respectively.

**136 2.1.3 Land-use-based agriculture sector GDP**

To better reflect the spatial structure of production activities, we introduce the supplier effect, which assumes a beneficiary-supplier relationship. Specifically, agricultural production occurring in peri-urban or rural areas surrounding major population centers is regarded as supplying food and resources to those urban beneficiaries. These agricultural zones, while themselves sparsely populated, are functionally integrated with the urban economy. Therefore, they are expected to exhibit higher

GDP values than similarly sparse regions that are not spatially or economically connected to urban demand. To capture this spatial interdependence, the supplier effect applies a distance-decay reallocation from beneficiary pixels in PB-method to nearby supply-side pixels, namely those identified as MCROP. Technically, this is implemented as a linear decay function, in which full weight is given within an inner threshold of 150 km, and weight decrease linearly to zero at an outer threshold of 300km.

**145**
$$w_{ij} = if d_{ij} \le d_{in}: 1; if d_{in} < d_{ij} \le d_{out}: 1 - (d_{ij} - d_{in})/d_{in}; if d_{ij} > d_{out}: 0$$
 (2)

| Sector      | Definition of ISIC |  |  |
|-------------|---------------------------|--|--|
| Agriculture | ISIC 01-03 (A)            |  |  |
| Service*    | ISIC 50-99                |  |  |
| Industry    | ISIC 05-43 (B-F)          |  |  |

\*Noted that only the Service sector is based on ISIC Rev. 3.

Table 2: Definition of each sector, based on the International Standard Industrial Classification (ISIC) Rev 4, in the GDP data by the World Bank (2023).

**150 2.1.4 Land-use-based service sector GDP**

Similarly, PB-method of the service sector is reallocated to residential areas (RES) by applying the supplier effect. The rationale here differs slightly from that for agriculture. Grid-scale population data (e.g., at 30 arcsec resolution, or approximately  $1 \times 1$  km per pixel) are too fine to represent realistic service usage, since people commonly travel more than 1 km by car or public transportation to access services (Ciccone and Hall, 1996). Therefore, this reallocation is designed to represent commuting patterns, where service activities in peri-urban zones support nearby urban demand centers. In this context, we use a supplier effect with an inner threshold of 25 km (representing high-intensity interaction) and an outer threshold of 50 km, beyond which service contributions are assumed negligible.

**158 2.1.5 Land-use-based industry sector GDP**

We distributed the industry sector GDP in each country by multiplying the distributed GDP per pixel by the NRES in each pixel.

Thus, the distribution was performed for each country, as follows:

**161** Industry GDP per pixel
$$_{country} = Total Industry GDP _{country} / \sum_{i=1}^{n} NRES_{i}$$
 (3)

Industry GDP
$$_{country}^{i} = Industry GDP per pixel_{country}^{i} \times NRES_{i}^{i}$$
 (4)

where is the Industry GDP per pixel of sector s in the country, is the total sectoral GDP of industry in the country, is the non-residential area in pixel i, n is the total number of pixels in the country, and is the distributed industry GDP in pixel i in the country.

**166 2.2 Comparison of GDP distribution methods**

We created two types of spatial distributed GDP map: population-based (PB-method), Land-use-based (LUB-method). The PB
map was generated by downscaling the country GDP only in proportion to the gridded population count into a 30 arcsec map. The
LUB-method was generated for each sectoral area and sectoral GDP per area. To assess the effectiveness of the proposed LUB
mapping approach, we compared it against PB-method using the DOSE dataset (Wenz et al., 2023), which provides sectoral GDP
estimates at the sub-national administrative unit level (GADM level 1). Both GDP maps (i.e., PB-method and LUB-method) were
spatially aggregated from 30 arcsec resolution to the corresponding GADM Level 1 administrative boundaries to enable direct
comparison with DOSE data. Comparison involved three steps: (1) Scatter plots were generated to evaluate the agreement
between the aggregated values from each GDP map and corresponding sectoral GDP values from the DOSE dataset (agriculture,
service, and industry) used as reference data. (2) For each method and sector, we computed the absolute value of the relative error
between estimated and reference GDP values and derived the cumulative distribution functions to illustrate the distribution of
errors across all administrative units. (3) We computed the difference in absolute relative errors between the LUB-method and
PB-method to evaluate the improvement or deterioration in accuracy. For each administrative unit, this metric was calculated as:

$$\Delta E = E_{LUB} - E_{PB}$$
, where  $E = \frac{\left|GDP_{estimate} - GDP_{DOSE}\right|}{GDP_{DOSE}}$  (5)

A negative value of ( $\Delta E$ ) indicates that LUB-method is closer to the reference than PB-method (i.e., an improvement), while a 181 positive value indicates a deterioration in accuracy compared to PB-method. The comparison was conducted using only 182 administrative units for which all three sectoral GDP values were available for the year 2010. In total, the comparison included 183 1,165 administrative units across 57 countries.

**184 3 Results**

We developed three GDP maps for service, industry, and agriculture sectors in 2010, 2015, and 2020. We excluded other years because of the low coverage of national GDP statistics in the World Bank data. Hereafter, the map generated using the LUB method within the Methods will be referred to as "SectGDP30", and the map generated using the PB method will be referred to as "PB-method". The maps of SectGDP30 are shown in Fig. 2 (a), (b), and (c). Additionally, to clarify the difference of spatial distribution among sectors, we showed (d) the map of the largest GDP sector in each grid in the world. Globally, the distribution of economic sectors generally correlates with population distribution, with concentrations observed in urban centers. However, variations exist in the detailed distributions. The service sector's distribution predominantly concentrates in urban areas across countries, consistent with population distribution patterns and the use of residential data. In contrast, industrial GDP, proxied by non-residential areas, shows a tendency toward greater concentration in coastal regions. Conversely, agricultural GDP, while exhibiting some correlation with population distribution, is characterized by a more expansive distribution in inland areas compared to the service sector.

Examining individual countries allows for the identification of more specific differences in the distribution of each sector at a finer 198 scale, shown in Fig. 3. In the figure of Japan, Japan's three major metropolitan areas—Tokyo, Osaka, and Aichi—shows variations in sectoral distribution, despite their common characteristic of high population concentration. In the GDP map, the service sector 200 predominates in the coastal areas of Tokyo and Osaka, which are marked by high population and service industry presence. In 201 contrast, Aichi's coastal regions exhibit a widespread predominance of industrial GDP. Industrial GDP is not uniformly 202 distributed across the entire Aichi area. Within Aichi, the more inland urban center, such as the Nagoya area, shows a prevalence 203 of the service sector, with industrial GDP concentrated in coastal areas. These findings align with Aichi's higher proportion of 204 industrial GDP compared to Tokyo and Osaka (Wenz et al., 2023<del>DOSE, 2024)</del>, and the formation of an extensive industrial belt 205 along its coastal regions. This dataset facilitates the depiction of detailed distributional differences within these areas.

When comparing central Bangkok with its southeastern region, a similar pattern emerges as a case in Japan. The southeastern 208 area, specifically the Eastern Seaboard and Eastern Economic Corridor (EEC) centered around Laem Chabang Port, has 209 developed as an industrial hub. In this region, industrial GDP predominates over service sector GDP. Regarding the distribution of 210 agricultural GDP, Japan shows fewer pixels where agricultural GDP is dominant, largely because much of its agricultural land is 211 located relatively close to urban areas. However, in Thailand and France, extensive areas with dominant agricultural GDP are 212 observed around metropolitan centers like Bangkok and Paris. For instance, Figure 4 (a), which shows only agricultural GDP for 213 France, illustrates that agricultural GDP is minimally developed around densely populated Paris. Conversely, it depicts 214 widespread agricultural activity in the less populated surrounding regions.

Figure 2: The sectoral GDP maps of (a) service sector, (b) industry sector, (c) agricultural sector, (d) the map of the largest GDP 218 sector in each grid of 30 arcsec.

Figure 3: The map of the largest GDP sector in each grid of 30 arcsec in (a) France, (b) Thailand, and (c) Japan.

To validate the accuracy of this GDP map, we conducted a comparative analysis with DOSE, a dataset providing sectoral GDP 224 figures at the sub-national administrative unit level. For this validation, the 30 arcsec resolution GDP map was spatially 225 aggregated according to the GADM dataset's Level 1 administrative divisions, which are used by DOSE. The aggregated GDP 226 values for each administrative unit were then calculated and compared with DOSE's figures.

The results are presented in Figure 4 (a), (b), and (c). These three scatter plots indicate that SectGDP30 exhibits a similar 229 distribution to actual sub-national scale sectoral GDP ( $R^2 > 0.9$  in all the sectors). When examined by sector, many administrative units with discrepancies in service and industrial GDP show an underestimation compared to actual data. Given that the total GDP per sector at the national level aligns with real data in this study, this discrepancy likely results from over-distributing GDP in a few administrative units within certain countries, leading to an underestimation in many other smaller administrative units. While service and industrial GDP inherently concentrate in specific local areas, and this GDP map depicts that, some countries show an excessive concentration in particular regions. This trend is less apparent in agricultural GDP, which exhibits less localized distribution, and no strong pattern of overestimation or underestimation was observed.

Next, we compared the results from SectGDP30 with PB-method. The comparison method involved using sectoral GDP figures for 238 each administrative unit, as before, and calculating the cumulative distribution of the differences from DOSE's figures. This result 239 is presented in Figure 4 (d). Sectoral analysis reveals that the industrial sector shows the most significant improvement when 240 compared to PB-method. As previously mentioned, industrial GDP distribution often exhibits localized concentrations even in 241 sparsely populated areas. This suggests that a method using only non-residential land use information and concentrating 242 distribution over relatively small areas is more appropriate than PB-method, which relies on population distribution data.

The service sector shows a slight decline in accuracy compared to PB-method. In the service sector, overall regional results showed 245 a slight decrease in accuracy for SectGDP30 compared to PB-method. However, some regions exhibited improved accuracy with 246 SectGDP30. Fundamentally, there is minimal difference between SectGDP30 and PB-method as the spatial distributions of 247 residential areas (upon which SectGDP30 relies) and population (upon which PB-method relies) largely coincide.

Conversely, SectGDP30 incorporates Supplier effect, reallocating each grid's GDP to residential areas within a 50km radius. This 249 results in a smoother connection of urban and rural area distribution differences compared to PB-method. This effect is evident in 250 the Alpine regions of Switzerland (CHE), specifically in administrative level districts such as Uri, Wallis, Graubunden, and Glarus. 251 While these Swiss Alpine areas have a significant population, residential areas are limited, and actual statistical service GDP is not 252 high. Therefore, in Switzerland, service GDP should be distributed not based on simple population distribution but rather in the 253 plains north of the Alps, where numerous residential areas exist. This case demonstrated an improvement in SectGDP30 accuracy. 254 Agricultural GDP also shows an improvement compared to PB-method, with an increase in the number of administrative units 255 exhibiting smaller errors.

Figure 4: The scatter graphs of the municipality GDP for (a) service sector (b) industry sector (c) agriculture sector and (d) the 259 cumulative distribution of the errors between DOSE and SectGDP30 and between DOSE and PB-method for each sector.

[revised manuscript text omitted]

BI loss [billion USD, current value in 2011]

Figure 5: Spatial distribution of the inundation period of the 2011 Thailand flood, obtained from (a) Catchment-based Macro-scale 306 Floodplain (CaMa-Flood) simulation and (b) Moderate Resolution Imaging Spectroradiometer (MODIS) observation data, and 307 the simulation Business interruption losses (USD billion, current value in 2011) due to the 2011 Thailand flood, estimated by 308 combining hazards and exposures; the total loss is written in the center of each circle. (c) CaMa-Flood and PB-method, (d) 309 CaMa-Flood and SectGDP30, (e) MODIS and PB-method, (f) MODIS and SectGDP30, and (g) the World Bank report (2011).

Firstly, comparing the losses by the different hazard data with the same exposure, SectGDP30, the service sector loss according to 312 CaMa-Flood (USD 15.86 billion) was over 12-fold larger than that according to MODIS (USD 1.29 billion). This large difference 313 was caused by the shorter average inundation period and smaller flood area in MODIS than in CaMa-Flood. MODIS is known to 314 tend to fail to capture the flood extent in urban areas with high densities of tall buildings and that leads to the underestimation in 315 inundation. In addition to different total losses, ratios of industry sector loss to the total loss differed between two results: 48.20% 316 according to CaMa-Flood and 35.62% according to MODIS. This result showed the sectoral ratio of the loss can be changed 317 depending on spatially different hazards. It is caused by the fact that SecGDP30 can show the different spatial distribution of each 318 sectoral GDP, while municipality-level statistics cannot show the spatial distribution in a fine resolution. This sectoral difference 319 was newly found by this study since the traditional population-based GDP map also could not show this difference between 320 sectors.

Comparing the results using CaMa-Flood and SectGDP30 with the World Bank Report figures (Figure 5 (d) and (g)), SectGDP30 323 more accurately represents the smaller proportions of agricultural damage compared to when PB-method is used (Figure 5 (c)). 324 This indicates that SectGDP30 can effectively constrain the allocation of agricultural GDP in areas with high population but 325 limited agricultural land. Conversely, while the Report figures show a significant proportion for the industry sector, SectGDP30 326 results estimate the industry sector to be almost on par with the service sector. It showed the industry loss was underestimated 327 although the hazard in the numerical simulation, by CaMa-Flood, captured the flood extent over the industrial sector area and the 328 long-lasting inundation period. The reported value excludes assets damage but includes economic losses other than production 329 reduction by direct contact with the flood, such as production stoppage due to shortages of raw materials induced by blocked 330 roads. Therefore, if we assume that the new sectoral GDP map captured the industrial locations and they were successfully 331 considered to be flooded, this underestimation is presumed to be caused by a lack of data reflecting the indirect production 332 stoppage.

Related to this limitation of the indirect production stoppage, it is important to recognize that the methodology, including that of 335 this paper and previous studies, which determines the GDP produced in each pixel using indicators such as GDP per unit area, 336 overlooks the fact that labor supplied from remote locations is necessary for GDP production. To rephrase this with the example of 337 a factory affected by a disaster: while the GDP output itself occurs at the factory's location, the workers who carry out the 338 production reside in surrounding or remote areas. Therefore, if a disaster occurs in these remote residential areas, the GDP output 339 should cease. However, pixel-based calculation methods would fail to represent this cessation of GDP output as long as the 340 factory's pixel is unaffected. This is considered a non-negligible impact in regions where economic activity and residential areas are 341 clearly separated, but quantifying this impact on a global scale is currently challenging. Alongside future research on regional 342 differences in GDP per unit area, this remains a limitation that we must consider moving forward.

**343 5 Data availability**

The global sectoral GDP maps are publicly available via Zenodo at 345 https://doi.org/10.5281/zenodo.15774017https://doi.org/10.5281/zenodo.13991673 (Shoji et al., 2025<del>2024</del>). The maps on 346 Zenodo correspond to the SBCE maps in this paper and are stored as geotiff files. In total, there are nine maps in the dataset, 347 for each sector (service, industry, and agriculture) and year (2010, 2015, and 2020).

**348 6 Summary**

This study developed a spatially distributed sectoral GDP map (SectGDP30) by leveraging recently available global, 350 high-resolution land use datasets. This map demonstrates strong consistency ( $R^2 > 0.9$ ) with actual sub-national statistical 351 data and exhibits greater alignment with sub-national GDP statistics compared to conventional GDP maps (PB-method) that 352 rely solely on gridded population maps.

For the industry sector, the methodology successfully distributed industrial GDP with better accuracy than population 355 distribution alone. This was achieved by adopting "Non-residential areas" as a proxy, which effectively captures the localized 356 nature of industrial GDP distribution in specific regions within each country. For agriculture, accuracy was improved over 357 PB-method by distributing GDP based on farmland maps and assuming GDP generation in areas approximately 150-300 km 358 from wide-area population centers. Regarding the service sector, incorporating population distribution within specific ranges, 359 even when using residential land use map information, resulted in GDP being distributed only to actual built-up and 360 designated residential areas. This approach achieved an accuracy comparable to PB-method.

As an application of this dataset, business interruption (BI) loss estimation due to floods was conducted using the sectoral GDP map. This confirmed that the new sectoral GDP map can represent inter-sectoral differences in estimated BI losses, 364 corresponding to varying spatial distributions of hazards. This validation underscores the importance of considering the 365 spatially distinct distributions of sectors when estimating actual disaster damage. It also highlights the need for developing 366 new estimation methods that account for the processes of GDP generation.

This new global sectoral GDP map serves as a foundational tool for estimating sector-classified economic losses. It 369 meticulously considers the complexity of global land use patterns at a detailed level, enabling accurate calculation of 370 sector-specific losses from various natural disasters on a global scale.

**372 Author contributions**

- 373 The SectGDP30 dataset was conceptualized by TS and DY. Data processing and validation were performed by KK and TS.
- 374 The application of the maps of the SectGDP30 in the case of the Thailand flood was performed by TS. The remaining
- 375 co-authors participated in the editing of the paper.

**377** Competing interests**

[revised manuscript text omitted]

- 463 Tanoue M, Hirabayashi Y, Ikeuchi H.: Global-scale river flood vulnerability in the last 50 years. Scientific Reports 6: 36021,
- **464** 2016.
- 465 Tanoue M, Taguchi R, Nakata S, Watanabe S, Fujimori S, Hirabayashi Y.: Estimation of direct and indirect economic losses
- 466 caused by a flood with long-lasting inundation: Application to the 2011 Thailand flood. Water Resources Research 56, 2020.
- 467 Tanoue M, Taguchi R, Alifu H, Hirabayashi Y.: Residual flood damage under intensive adaptation. Nature Climate Change
- 468 11: 823-826, 2021.
- 469 Tellman B, Sullivan JA, Kuhn C, Kettner AJ, Doyle CS, Brakenridge GR, Erickson TA, Slayback DA.: Satellite imaging
- 470 reveals increased proportion of population exposed to floods. Nature 596: 80–86, 2021.
- 471 The European Environmental Agency.: CORINE Land Cover.
- 472 https://land.copernicus.eu/en/products/corine-land-cover?tab=main, 2017.
- 473 Theobald DM.: Development and Applications of a Comprehensive Land Use Classification and Map for the US. PLoS
- 474 ONE 9: e94628, 2014.
- 475 Wenz L, Carr RD, Kögel N, Kotz M, Kalkuhl M.: DOSE Global data set of reported sub-national economic output. Sci
- 476 Data 10: 425, 2023.
- 477 Wenz L, Willner SN.: 18. Climate impacts and global supply chains: An overview. Handbook on Trade Policy and Climate
- 478 Change, 290, 2022.
- 479 Willner SN, Otto C, Levermann A.: Global economic response to river floods. Nature Climate Change 8: 594–98, 2018.
- 480 The World Bank.: 2011 Thailand Floods: Rapid Assessment for Resilient Recovery and Reconstruction Planning.
- 481 https://recovery.preventionweb.net/publication/2011-thailand-floods-rapid-assessment-resilient-recovery-and-reconstruction-
- **482** planning, 2011.
- 483 The World Bank.: World Development Indicators. https://databank.worldbank.org/source/world-development-indicators,
- **484** 2023.
- 485 Yamazaki, D, Kanae S, Kim H, Oki T.: A physically-based description of floodplain inundation dynamics in a global river
- 486 routing model: FLOODPLAIN INUNDATION DYNAMICS. Water Resources Research 47: w04501, 2011.
- 487 Zhuang, L., & Ye, C.: More sprawl than agglomeration: The multi-scale spatial patterns and industrial characteristics of
- 488 varied development zones in China. Cities, 140, 104406, 2023